# Persistent homology elucidates hierarchical structures responsible for mechanical properties in covalent amorphous solids

Emi Minamitani [1,2] ✉, Takenobu Nakamura[3], Ippei Obayashi[4] & Hideyuki Mizuno [5]

Understanding how atomic-level structures govern the mechanical properties of amorphous materials remains a fundamental challenge in solid-state physics. Under mechanical loading, amorphous materials exhibit simple affine and spatially inhomogeneous nonaffine displacements that contribute to the elastic modulus through the Born (affine) and nonaffine terms, respectively. The differences between soft local structures characterized by small Born terms or large nonaffine displacements have yet to be elucidated. This challenge is particularly complex in covalent amorphous materials such as silicon, where the medium-range order (MRO) plays a crucial role in the network structure. To address these issues, we combined molecular dynamics simulations with persistent homology analysis. Our results reveal that local structures with small Born terms are governed by short-range characteristics, whereas those with large nonaffine displacements exhibit hierarchical structures in which short-range disorder is embedded within the MRO. These hierarchical structures are also strongly correlated with low-energy localized vibrational excitations. Our findings demonstrate that the mechanical responses and dynamic properties of covalent amorphous materials are intrinsically linked to the MRO, providing a framework for understanding and tailoring their properties.

The mechanical responses of amorphous materials have long been the subject of extensive research[1]. Crystalline materials respond to strain through defects and dislocations, but amorphous materials, with disordered and nonperiodic structures, do not have clearly identifiable response mechanisms. Several theoretical models[2,3], including shear transformation zone theory[4,5], have been proposed to describe their mechanical behavior. However, these models assume the presence of structural defects without providing precise definitions, making the establishment of a direct link between the local atomic structures and mechanical properties challenging. This limitation has led to an ongoing debate on the role of specific structural motifs in amorphous materials, such as free volume and bond orientation in metallic glasses[6-9].

Covalent amorphous materials, including amorphous silicon (a-Si) and silicate glasses, present an even greater challenge for understanding structure–property correlations because these materials have complex network structures. Previous studies introduced various structural descriptors to characterize local environments, such as interatomic angle distributions[10,11], flexibility volume[12,13], and average coordination numbers motivated by topological constraint theory[14,15].

[1]SANKEN, The University of Osaka, 8-1 Mihogaoka, Ibaraki, Osaka 567-0047, Japan. [2]JST, PRESTO, 4-1-8 Honcho, Kawaguchi, Saitama 332-0012, Japan. [3]Department of Materials and Chemistry Materials DX Research Center, National Institute of Advanced Industrial Science and Technology (AIST), 1-1-1 Umezono, Tsukuba, Ibaraki 305-8568, Japan. [4]Center for Artificial Intelligence and Mathematical Data Science, Okayama University, Okayama 700-8530, Japan. [5]Graduate School of Arts and Sciences, The University of Tokyo, Tokyo 153-8902, Japan. ✉e-mail: eminamitani@sanken.osaka-u.ac.jp

These indicators have been correlated with mechanical properties such as shear modulus and hardness, suggesting that atomic-scale structural features influence mechanical behavior. Despite these insights, a clear connection between elasticity and nanostructure remains elusive, and two major issues remain unresolved.

The first problem is the inhomogeneity of the atomic displacement in amorphous materials. Under an applied strain, atomic displacements consist of affine components, which follow the overall strain, and nonaffine components, which deviate from it and are spatially inhomogeneous (Fig. 1a). These two types of deformations contribute differently to the elastic modulus[16–19]. The contribution from the affine displacements is known as the Born term, whereas that from the nonaffine displacements is referred to as the nonaffine term. The overall elastic modulus is obtained by subtracting the nonaffine term from the Born term. Consequently, soft structures that readily respond to strain and lower the elastic modulus are regions with a small Born term and/or large nonaffine displacement. The spatial distribution of softness, which originates from nonaffine displacement, is highly inhomogeneous[20,21], leading to characteristic atomic dynamics such as the boson peak anomaly[22,23] and low-energy localized vibrations[24,25]. This inhomogeneity complicates efforts to correlate softness and structure using traditional mean-field descriptors such as angle distributions and average coordination numbers.

In addition to the inhomogeneous atomic displacement, another key aspect is the interplay between softness and the medium-range order (MRO), which is a distinctive feature of amorphous structures. Generally, MRO refers to atomic organization on a scale of 5–20 Å, which does not appear in purely random configurations[26,27]. In covalent amorphous materials, network structures consist of polyhedral units created by atomic bonds that share corners, edges, and faces. Figure 1b shows an example of a corner-sharing network. The organization of the bond-length scale within polyhedral units is called the short-range order (SRO), whereas the MRO corresponds to the relative angles and connectivity between polyhedra[26,27]. The angle between two adjacent polyhedra corresponds to the dihedral angle formed by four atoms, the restriction of which corresponds to the smallest MRO scale. When the angles between multiple polyhedra are correlated, the atomic ring structures are constrained and an MRO is imposed on a longer scale.

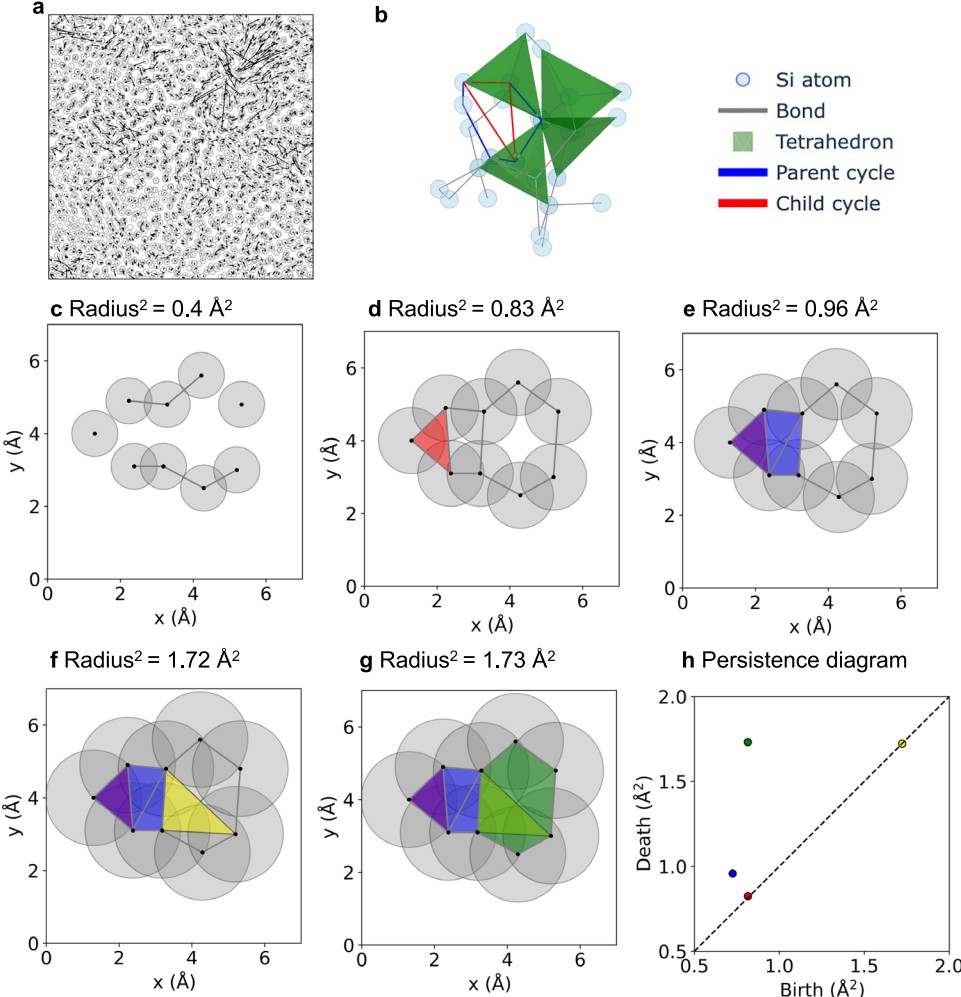

**Fig. 1 | Schematics of the key concepts and method of persistent homology used in this study. a** Visualization of nonaffine displacement in a three-dimensional amorphous structure under shear deformation, with a slice in the *xy*-plane. Gray circles denote atomic positions, and black arrows represent nonaffine displacements projected onto the *xy*-plane. The model system consists of 13,824 Si atoms in a cubic unit cell with a side length of 65 Å. The displayed slice corresponds to atoms with z-coordinates between 40 Å and 50 Å. **b** Illustration of a typical covalent amorphous network structure using a-Si as an example. SiSi$_4$ tetrahedral units share corners and MRO defined by the angles between these tetrahedra. Blue and red polygons show examples of a parent cycle and its associated child, respectively, observed in a covalent amorphous network. These cycles represent the hierarchical nature of the MRO at different scales. **c–g** Filtration process in persistent homology with increasing sphere radii. Filled polygons represent cycles transforming into boundaries. Sequence of empty and filled polygons depicts the concept of children: the red triangle is a child of the blue pentagon, and the yellow triangle is a child of the green hexagon. **h** Persistence diagram obtained from the filtration process. Squared radii (Å²) are used as birth and death radii. Colors correspond to the polygons in the filtration process.

**a**

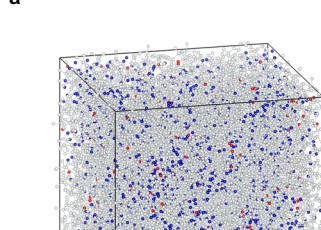

**b**

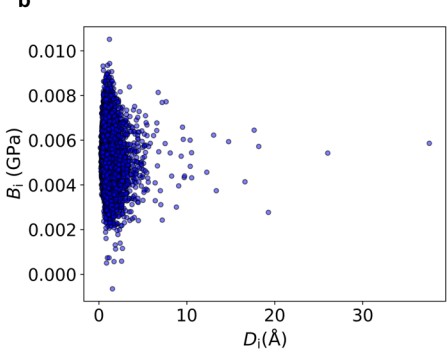

**Fig. 2 | Structural model of amorphous Si and calculated Born term ($B_i$) and nonaffine displacement ($D_i$) for each atom in the model. a** Amorphous structure comprising 13,824 Si atoms used for the analysis. White atoms represent the 4-coordinated Si atoms, while blue and red atoms indicate the Si atoms with coordination numbers of $\geq 5$ and $\leq 3$, respectively. **b** Scatter plot of $D_i$ and $B_i$. Source data are provided as a Source Data file.

However, the relationship between the MRO in these covalent amorphous networks and softness remains unexplored.

Motivated by these challenges, the present study aims to clarify the correlation between the MRO, nonaffine displacements, and low-energy localized vibrations based on persistent homology[28,29]. Persistent homology offers the advantage of simultaneously capturing structural features, such as rings and voids, at various scales. Consequently, persistent homology has been used to analyze the structures and their correlations with material properties in disordered materials[30-41]. Although persistent homology cannot describe fine differences in specific ordered structures such as fcc and hcp crystals with the same interatomic distances, this limitation is less important in disordered systems.

The key idea in persistent homology is to track the evolution of atomic connectivity as a function of the spatial scale (Fig. 1c–g). Imagine placing a sphere at the position of each atom and gradually increasing its radius. When two spheres touch each other, a bond is formed between the atoms. As the radius increases, the number of bonds increases, and polygonal rings are formed at a certain point. These rings correspond to one-dimensional holes in the topological space and are referred to as cycles. As the sphere radius continues to increase, the polygons become covered with the spheres. At this point, the cycle is transformed into the boundary of a filled polygon, indicating that the hole no longer exists. The structural features are described by the birth and death radii of these cycles, which represent the radii at which a cycle appears (birth radius) and transitions to a boundary (death radius), respectively. These pairs are visualized in a scatter plot called a persistence diagram (Fig. 1h).

Hierarchical information corresponding to the MRO is embedded in the persistence diagram. As a cycle transitions to a boundary, smaller cycles may form and subsequently transition to boundaries within a larger cycle (red triangle in Fig. 1d). In this study, we refer to such small inner cycles as children, whereas in other reports, this concept is referred to as a secondary ring[30].

The presence of children indicates that a smaller-scale order exists within a larger-scale order, which reflects the multiple scales of the MRO observed in covalent amorphous materials. The blue and red polygons in Fig. 1b provide examples of this correspondence. The red triangle representing the child cycle is associated with only two tetrahedra and reflects the smallest-scale MRO. By contrast, the blue hexagon, which represents the parent cycle, is associated with three tetrahedra and corresponds to a larger-scale MRO. This demonstrates the mechanism of capture of the multiscale nature of the MRO by the hierarchical structure in the persistence diagram.

In this study, we fully utilized the advantages of persistent homology to investigate the correlation between the local structure and mechanical responses in covalent amorphous materials under shear deformation. The proposed method can be broadly applied to various amorphous covalent materials. However, we used the prototypical model system of a-Si as a representative case because of its well-characterized tetrahedral network structure and single-element composition, which simplifies the persistent homology analysis and extensive experimental and theoretical studies[10,13,42-49].

Our findings reveal distinct characteristics of two types of soft local structures: regions with small Born terms and regions with large nonaffine displacements, which are linked to the SRO and MRO, respectively. Regions surrounding atoms with small Born terms consist of small rings with relatively few vertices and no children and are characterized by disorder in the bond-length and bond-angle distributions—a hallmark of disrupted SRO. By contrast, the regions surrounding atoms with large nonaffine displacements are characterized by larger rings with more vertices; these rings contain children and form hierarchical structures that reflect the multiscale MRO. Furthermore, atoms with large nonaffine displacements are associated with large amplitudes of low-energy localized vibrations, suggesting that the constraints imposed by the MRO are critical in determining the dynamical and mechanical responses. This insight provides a perspective for understanding and engineering the physical properties of covalent amorphous materials based on their static structures.

## Results

### Born term and nonaffine displacement

First, we examined the correlation between the Born term and nonaffine displacement under shear strain. The a-Si structure used in this study is illustrated in Fig. 2a. In this study, we use the Stillinger–Weber (SW) potential[50]. Although this potential tends to overestimate the population of fivefold coordinated atoms in a-Si[49], we employed it as a representative model to investigate the structural features governing the mechanical responses using persistent homology.

Figure 2b shows the scatter plot of the Born term, $C_{i,\alpha\beta\kappa\chi}^{\mathrm{Born}}$, and nonaffine displacement, $\mathbf{D}_{i,\alpha\beta}^{\mathrm{NA}}$, for each atom in the a-Si structure depicted in Fig. 2a. The Greek subscripts of $C_{i,\alpha\beta\kappa\chi}^{\mathrm{Born}}$ and $\mathbf{D}_{i,\alpha\beta}^{\mathrm{NA}}$ specify the direction, $x$, $y$, or $z$. The detailed definitions of $C_{i,\alpha\beta\kappa\chi}^{\mathrm{Born}}$ and $\mathbf{D}_{i,\alpha\beta}^{\mathrm{NA}}$ are provided in the Methods section. For the Born term, we used the calculated value of $B_i = (C_{i,xyxy}^{\mathrm{Born}} + C_{i,yzyz}^{\mathrm{Born}} + C_{i,zxzx}^{\mathrm{Born}})/3$; for the nonaffine displacement, we used the calculated value of $D_i = (|\mathbf{D}_{i,xy}^{\mathrm{NA}}| + |\mathbf{D}_{i,yz}^{\mathrm{NA}}| + |\mathbf{D}_{i,zx}^{\mathrm{NA}}|)/3$.

As depicted in Fig. 2b, no clear correlation exists between the Born term and nonaffine displacement. In particular, atoms with large nonaffine displacements are generally not associated with small Born terms. This result is also supported by the spatial distributions of the Born term and nonaffine displacement, as shown in Supplementary Fig. 1. This finding suggests that soft local structures characterized by small Born terms differ from those associated with large nonaffine

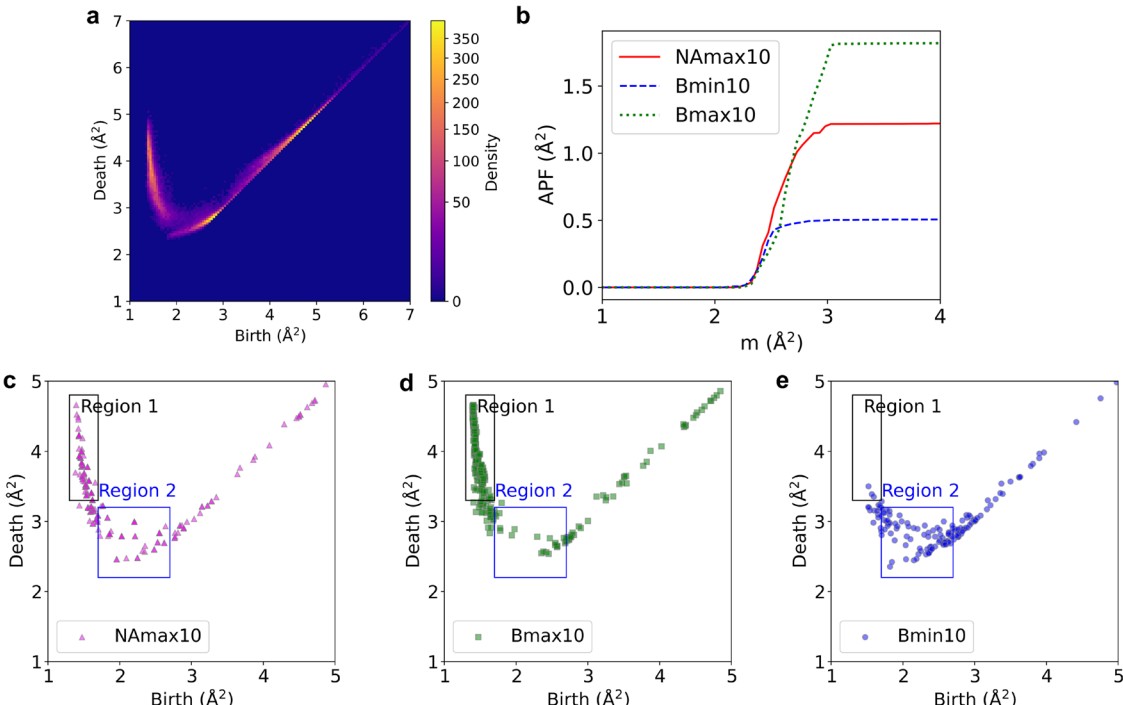

**Fig. 3 | Persistence diagram and birth–death pair distributions. a** Persistence diagram computed from the structure shown in Fig. 2a, visualized as a two-dimensional histogram. The colormap indicates the local density of birth–death pairs. **b** APF for the birth–death pairs assigned to respective groups of atoms (Bmax10, Bmin10, and NAmax10). A birth–death pair is classified into one of these groups or remains unclassified, depending on whether the corresponding ring structure, obtained through inverse analysis, contains at least one atom from the respective group. Source data are provided as a Source Data file. **c–e** Distribution of birth–death pairs for NAmax10 (**c**), Bax10 (**d**), and Bmin10 (**e**). Black square indicates the range of region 1, and blue square indicates the range of region 2. Source data are provided as a Source Data files.

displacements. To further explore the structural origins of mechanical softness, we analyzed three distinct atomic groups: Bmax10 (10 atoms with the largest Born terms), Bmin10 (10 atoms with the smallest Born terms), and NAmax10 (10 atoms with the largest nonaffine displacements).

## Differences in persistence diagrams

The distribution of birth–death pairs in a persistence diagram provides key insights into the structural differences associated with the Born term and nonaffine displacement. Figure 3a shows a persistence diagram obtained using the atomic coordinates of the sample shown in Fig. 2a. In Fig. 3a, owing to the large number of birth–death pairs, the data are visualized as a two-dimensional histogram.

In persistent homology, we can assign certain atomic ring structures to the respective birth–death pairs in a process called inverse analysis[51–53]. We analyzed the ring structures corresponding to all birth–death pairs within the interval [0, 5] for the birth and death radii. We then selected ring structures that included at least one atom from the corresponding groups: Bmax10, Bmin10, or NAmax10. For example, if a ring structure contains an atom from NAmax10, it is classified into the NAmax10 group. The scatter plots of the birth–death pairs for the selected rings in each of these groups are shown in Fig. 3c–e.

The distribution of birth–death pairs varies significantly between the Bmin10 and NAmax10 groups, clearly demonstrating that the structural features captured by persistent homology are distinct. To quantitatively evaluate the differences in birth–death pair distributions, we employed the accumulated persistence function (APF)[32], defined as follows:

$$\mathrm{APF}_G(m) = \frac{1}{|G|} \sum_{i:m_i < m} (d_i - b_i) \qquad (1)$$

Here, subscript $G$ identifies the NAmax10, Bmin10, and Bmax10 groups. $|G|$ is the number of birth–death pairs in each group. $b_i$ and $d_i$ are the birth and death radii for the $i$-th birth–death pair, and $m_i = \frac{d_i - b_i}{2}$.

The APF results highlight the differences in birth–death pair distributions among the Bmin10, Bmax10, and NAmax10 groups. All three APFs exhibit a step-like shape, with an initial rise near $m = 2.3$ Å² and saturation around $m = 3.0$ Å². Although the regions where the APF changes are the same among the groups, the saturation values differ markedly. Despite minor sample-to-sample variations, the saturated APF values consistently follow the order Bmax10 > NAmax10 > Bmin10. This result implies that the minimum difference between the death and birth radii is consistent among the groups. However, the numbers of pairs with a large difference, that is, pairs far from the diagonal, vary in this order.

This observation aligns with the trends observed in the scatter plots of birth–death pairs (Fig. 3c–e). For the Bmax10 and NAmax10 groups, birth–death pairs are primarily found in region 1 ([1.3, 1.7] and [3.3, 4.8] for the birth and death radii, respectively). By contrast, region 2 ([1.7, 2.2] and [2.7, 3.2]) contains fewer pairs. The Bmin10 group shows the opposite trend. These trends remain consistent regardless of whether we analyze the top 20 or top 50 atoms or examine 10 different samples (Supplementary Figs. 2 and 3). Thus, we conclude that the local structures of Bmax10 and NAmax10 are characterized by birth–death pairs in region 1, whereas Bmin10 is characterized by those in region 2.

## Differences in local structures

To gain deeper insight into the local structural differences between the Bmax10, NAmax10, and Bmin10 groups, we analyzed their associated atomic ring structures. Figure 4 shows the ring structures, including the atoms with the largest Born term (Bmax), largest nonaffine displacement (NAmax), and smallest Born term (Bmin). Rings containing

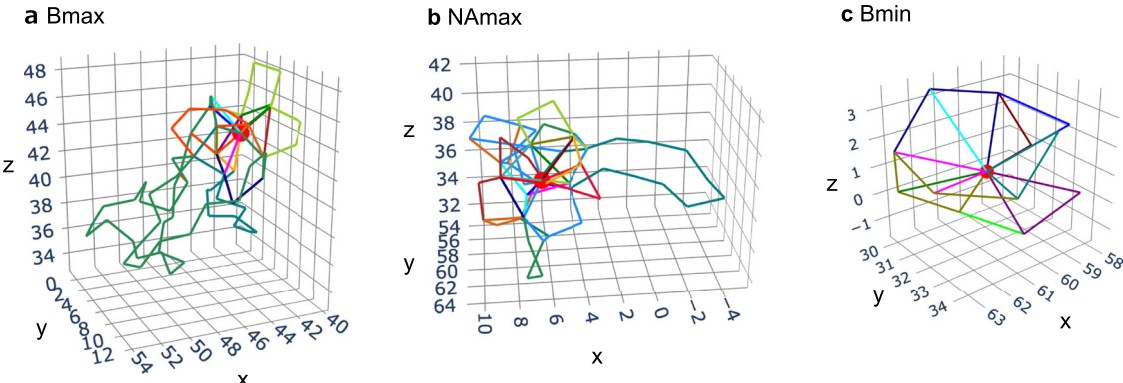

**Fig. 4 | Visualization of the ring structures determined by inverse analysis.** **a** Rings including the atom with the largest Born term (Bmax), **b** rings including the atom with the largest nonaffine displacement (NAmax), and **c** rings including the atom with the smallest Born term (Bmin). Position of each atom is marked by a red sphere. The color of the polygons denotes the corresponding cycle.

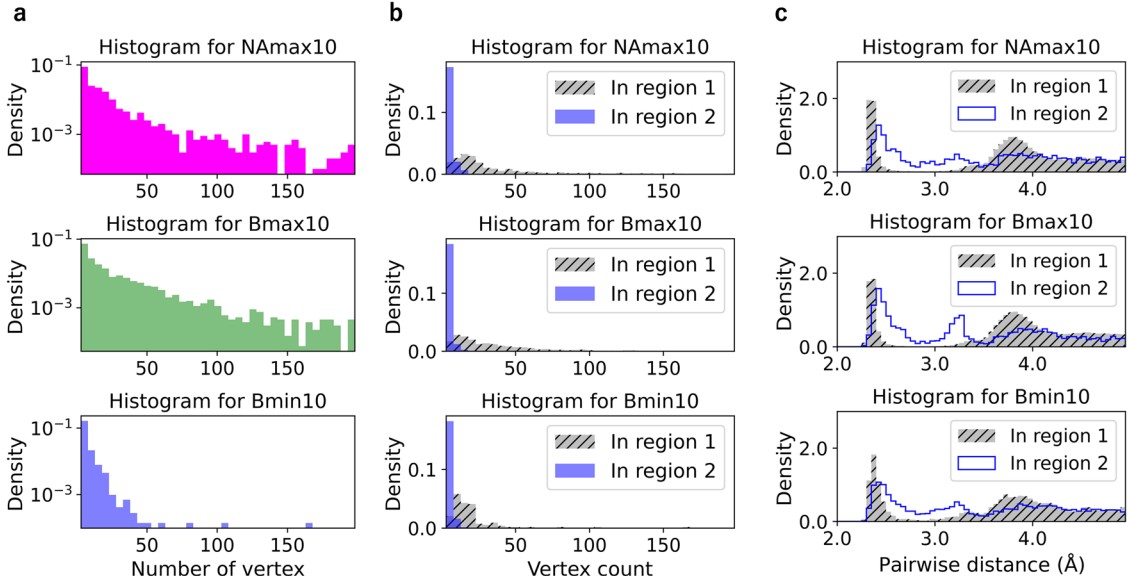

**Fig. 5 | Analysis of vertex counts and pairwise distances in ring structures.** **a** Histograms showing the vertex counts in ring structures for the NAmax10, Bmax10, and Bmin10 groups based on data from 10 samples. **b** Vertex count histograms for rings analyzed separately based on whether their birth–death pairs are in region 1 or 2. **c** Histograms of pairwise distances calculated using the atomic positions in the rings whose birth–death pairs are in region 1 or 2. To facilitate comparison between systems with varying atom counts, the histograms are normalized to probability density.

atoms with large Born terms and large nonaffine displacements exhibit numerous vertices and have larger sizes, corresponding to the birth–death pairs with large death radii. By contrast, those containing atoms with small Born terms tend to be composed of small triangles.

To validate the generality of these findings, we identified ring structures in the NAmax10, Bmax10, and Bmin10 groups across 10 samples and constructed histograms of their vertex counts (Fig. 5a). The vertex count reaches 700; however, the range shown in Fig. 5 is limited to 3–200. Rings in the Bmin10 group tend to have substantially fewer vertices than those in the NAmax10 and Bmax10 groups. Similar trends persist regardless of whether we analyze the top 20 or top 50 atoms (Supplementary Fig. 4).

We also analyzed the vertex counts separately for ring structures whose birth–death pair positions lie in regions 1 and 2 (Fig. 5b). As expected from the differences in death radii values, the vertex counts for the rings whose birth–death pairs lie in region 2 are small, whereas the vertex counts for those in region 1 exhibit a long-tail distribution. This trend has also been observed in previous studies[30,32,33].

In addition to this simple difference in the ring size, the rings in these two regions also make distinct contributions to the MRO, as

shown in the histogram of pairwise interatomic distances. The histogram of pairwise distances between atoms forming ring structures whose birth–death pair positions lie in region 1 reveal a sharp first peak and a clear second peak near 3.8 Å, which correspond to the small-scale MRO (Fig. 5c). The position of this second peak aligns with the death radius distribution in region 1, estimated as follows: when the death radius is 4.0 Å$^2$, the radius of spheres in the filtration is 2.0 Å, leading to an interatomic pairwise distance of 4.0 Å. By contrast, the histogram for region 2 shows a broader first peak and an additional peak near 3.2 Å, which is absent in region 1. The presence of this additional peak indicates short-range disorder due to variation in bond lengths, which also results in bond angle variations (Supplementary Fig. 5) and prevents the formation of MRO.

Based on these results, we conclude that the birth–death pairs in region 2 correspond to structures with short-range disorder, whereas those in region 1 correspond to MRO structures. Similar distributions of birth–death pairs and their correlations with short-range disorders and MRO formation have been previously reported[36].

Statistical analysis of these 10 samples reveals that 61.7% and 12.0% of the cycles for Bmax10 contain birth–death pairs in regions 1

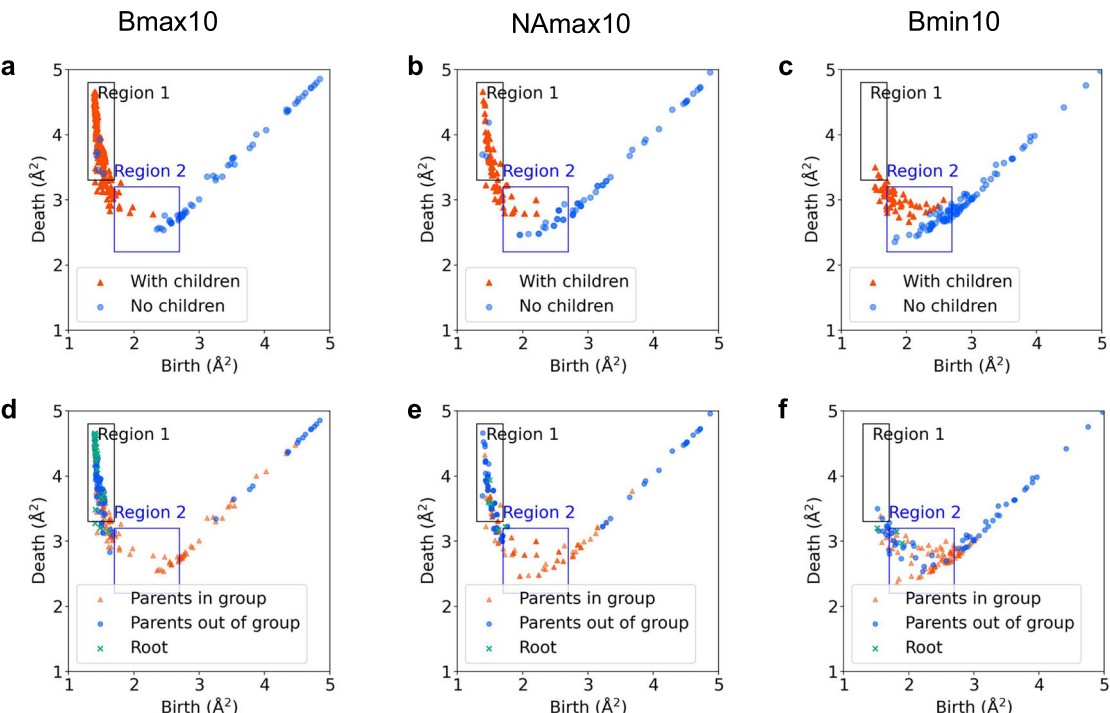

**Fig. 6 | Results of the child analyses. a–c** Presence or absence of children in the birth–death pairs assigned to the Bmax10 (**a**), NAmax10 (**b**), and Bmin10 (**c**) groups. **d–f** Presence or absence of parental birth–death pairs in the same group for the Bmax10 (**d**), NAmax10 (**e**), and Bmin10 (**f**) groups. Green crosses (roots) indicate the birth–death pairs without parents. Source data are provided as a Source Data files.

and 2, respectively. For NAmax10, the proportions are 51.1% and 17.2%, while those for Bmin10 are 7.4% and 45.1%, respectively. This indicates that the local environment surrounding the atoms associated with large Born terms has a lower short-range disorder and well-developed MRO. Conversely, the local environment surrounding the atoms associated with small Born terms exhibits greater short-range disorder and suppressed MRO. The local environment surrounding atoms with large nonaffine displacements closely resembles that of atoms with large Born terms. However, the larger ratio of birth–death pairs in region 2 indicates that these atoms exhibit unique local structures in which short-range disorder and MRO coexist.

These characteristics cannot be adequately captured by conventional local structural descriptors, such as the Voronoi volume and interatomic angle distributions. The Voronoi volume of each atom is not correlated with the nonaffine displacement and is only weakly correlated with the Born term (Supplementary Fig. 6). In the scatter plot of the average and standard deviations of the interatomic angles, which was proposed by Demkowicz and Argon as an indicator of the atomic environment[11], Bmax10 is concentrated in specific regions (green dots in Supplementary Fig. 6c). Therefore, the characteristics of the atoms associated with large Born terms are captured well using this indicator. By contrast, atoms in the NAmax10 and Bmin10 groups are distributed across various locations and exhibit no clear structural features. The correlation between the local environments of atoms with large nonaffine displacements and MRO, characterized by hierarchical structures, can only be elucidated via the application of persistent homology for multiscale analysis, as demonstrated in this study.

**Analysis of children**
The hierarchical structural characteristics associated with the MRO were further examined using the concept of children with persistent homology. This concept enables the quantitative evaluation of the nesting of smaller rings within larger ones, providing insight into the local structural motifs surrounding atoms with large Born terms or nonaffine displacements (Fig. 4a, b).

Figure 6a–c shows the distributions of birth–death pairs assigned to the Bmax10, NAmax10, and Bmin10 groups based on the presence or absence of children. In the Bmax10 and NAmax10 groups, birth–death pairs in region 1 contain children. Furthermore, Fig. 6d, e illustrates that in the Bmax10 and NAmax10 groups, birth–death pairs in region 2 are children of pairs with larger death radii within the same group. Notably, in the NAmax10 group, a non-negligible number of birth–death pairs in region 2 suggests that the local structures surrounding atoms with large nonaffine displacements exhibit a hierarchical structure corresponding to the MRO involving short-range disorder. By contrast, no hierarchical structure linking the short-range disorder with the MRO is observed for birth–death pairs in the Bmin10 group (Fig. 6f).

To validate the generality of these findings, we applied a similar analysis to 10 samples and plotted histograms of the number of children per birth–death pair in regions 1 and 2 separately (Fig. 7). Similar to the trends observed in the vertex counts, the number of children in the NAmax10 and Bmax10 groups exhibits long-tailed distributions and the Bmin10 group has fewer children. Moreover, the number of children per birth–death pair in region 2 is substantially lower than that in region 1. In the NAmax10 group, statistical analysis reveals that 91.9% of birth–death pairs in region 2 are children of other birth–death pairs within the same group, strongly indicating the presence of a hierarchical structure.

**Correlation with low-energy localized vibrational excitations**
The observed correlation between the hierarchical structure and nonaffine displacement is expected to influence the low-energy vibrational modes. The nonaffine displacement can be decomposed by the vibrational mode eigenvector, $\boldsymbol{\psi}_p$[16,54].

$$\mathbf{D}_{i,\alpha\beta}^{\mathrm{NA}} = \frac{\partial \mathbf{r}_i}{\partial \eta_{\alpha\beta}} = \frac{1}{\lambda_p} \sum_{j \neq i} \sum_p \left( \frac{\partial^2 U}{\partial \mathbf{r}_j \partial \eta_{\alpha\beta}} \cdot \boldsymbol{\psi}_p \right) \boldsymbol{\psi}_p \tag{2}$$

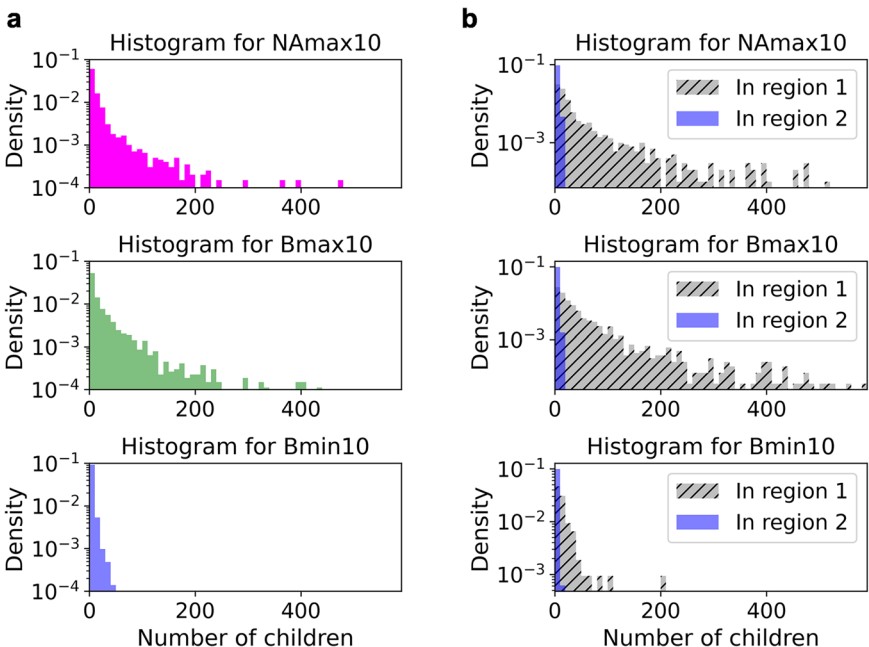

**Fig. 7 | Statistical analyses for number of children in 10 samples. a** Histograms of the number of children for the birth–death pairs assigned to the NAmax10, Bmax10, or Bmin10 groups. **b** Histograms of the number of children analyzed separately based on whether the birth–death pairs are in region 1 or 2.

Here, $U$ is the potential energy of the system, $\eta_{\alpha\beta}$ is the strain, $\alpha$ and $\beta$ are the indices specifying direction $x, y,$ or $z$, and $i$ and $j$ are indices specifying the atoms. $\lambda_p$ is the $p$-th eigenvalue of the Hessian matrix, and its unit is eV/Å².

Equation (2) suggests a strong correlation between the nonaffine displacement and low-energy localized vibrational modes, which can be attributed to the small value of $\lambda_p$ in the denominator and the finite value of $\frac{\partial^2 U}{\partial \mathbf{r}_j \partial \eta_{\alpha\beta}} \cdot \boldsymbol{\psi}_p$ in the numerator. In other words, the presence of low-energy localized vibrational modes is associated with a characteristic local structure in which the short-range disorder is embedded within the MRO. To verify this numerically, we analyzed the correlations between the amplitude of the localized modes, nonaffine displacement, and Born term. Additionally, we examined whether the atomic rings associated with Bmax10, Bmin10, and NAmax10 spatially overlapped with regions of high amplitude in the low-energy localized modes.

As a criterion for localization, we evaluated the participation ratio (PR), defined as follows:

$$\text{PR}(p) = \frac{1}{N_{atom}} \frac{\left(\sum_{i=1}^{N_{atom}} \left(\psi_{p_{ix}}\right)^2 + \left(\psi_{p_{iy}}\right)^2 + \left(\psi_{p_{iz}}\right)^2\right)^2}{\sum_{i=1}^{N_{atom}} \left(\left(\psi_{p_{ix}}\right)^2 + \left(\psi_{p_{iy}}\right)^2 + \left(\psi_{p_{iz}}\right)^2\right)^2}, \quad (3)$$

where $\psi_{p_{ix}}$, $\psi_{p_{iy}}$, and $\psi_{p_{iz}}$ denote the $x, y,$ and $z$ components of the $p$-th eigenvector, $\boldsymbol{\psi}_p$, for atom $i$. We also evaluated the inverse participation ratio (IPR) and PR for all the vibrational modes, and the results are summarized in Supplementary Fig. 7.

Lower PR values indicate stronger localization. However, the finite system size affects spatially localized vibrational modes because of hybridization with extended modes, which makes the threshold for separating the localized and extended modes ambiguous. To mitigate the hybridization and finite-system-size effects, we employed a well-established mode-demixing procedure[53,54]. Details regarding the mode-mixing procedure and its impact on the finite-system-size effect are presented in the Supplementary Information. Figure 8a shows the original and demixed PR values for the 1000 lowest-frequency modes. The results clearly show the separation of the localized modes below

2.5 THz. We identified low-energy localized modes with demixed PR values below 0.1 and computed their amplitudes for each atom using their eigenvectors as follows:

$$AMP_i = \sum_{p \in loc} \frac{1}{\lambda'_p} \left( \left| \psi'_{p_{ix}} \right| + \left| \psi'_{p_{iy}} \right| + \left| \psi'_{p_{iz}} \right| \right). \quad (4)$$

Subscript $p \in loc$ in Eq. (4) indicates the summation over the low-energy localized modes. $\lambda'_p$ is the updated $p$-th eigenvalue used during the demixing procedure, and its unit is eV/Å². $\psi'_{p_{ix}}$, $\psi'_{p_{iy}}$, and $\psi'_{p_{iz}}$ denote the $x, y,$ and $z$ components of the $p$-th demixed eigenvector. Although the nonaffine displacement of each atom is strongly correlated with $AMP_i$, no such correlation is observed with the Born term (Fig. 8b, c). This trend is preserved even when $AMP_i$ is calculated using the original eigenvectors (Supplementary Fig. 10). Additionally, the distribution of $AMP_i$ within the ring structures assigned to the Bmax10, Bmin10, and NAmax10 groups demonstrates that only the rings in the NAmax10 group have a strong spatial overlap with regions of large amplitudes of the low-energy localized modes (Fig. 8d). The correlations of particle-based IPR with nonaffine displacement and the Born term were also examined (Supplementary Fig. 11), leading to a similar conclusion.

Based on our results, we conclude that the hierarchical structure characterizing the local environment surrounding NAmax10, where short-range disorder is embedded within the MRO, facilitates the emergence of low-energy localized modes, leading to large nonaffine displacements.

## Discussion

By analyzing the shear modulus of a-Si, we demonstrated that the characteristics of soft local structures with small Born terms differ significantly from those with large nonaffine displacements. The local structures surrounding atoms with small Born terms are governed by the short-range characteristics, whereas those with large nonaffine displacements form hierarchical structures that reflect multiscale MRO. These MRO structures are also associated with large amplitudes of the low-energy localized vibrations. Because large nonaffine

displacements correspond to soft regions with low local elastic moduli[20,21], the MRO beyond the SRO scale plays a crucial role in influencing the dynamics and mechanical responses of amorphous materials. Such insights are challenging to obtain using conventional structural indicators such as the Voronoi volume or bond orientation. A comparison with indicators derived from recently proposed machine-learning-based models[55–57] could provide valuable directions for future research.

Our persistent homology analysis elucidates the boson peak and its structural origin. In glasses, excess vibrational states exist in the low-frequency regime beyond the Debye density of states of the crystals. This excess density of states, referred to as the boson peak, induces anomalous thermal properties such as excess specific heat and unusually low thermal conductivity[58]. Previous studies have consistently demonstrated a strong correlation between the localized vibrational modes and boson peaks. Shimada et al.[59] and Wyart et al.[60] demonstrated that localized modes originate from the boson peak and that their frequencies are reduced by repulsive interactions. Furthermore, Moriel et al.[61] revealed that the boson peak comprises multiple spatially coupled localized modes. Hu and Tanaka[62] reached similar conclusions. These studies indicate that the localized modes and boson peaks share a common origin. By integrating these insights with the correlation between the localized modes and MRO revealed in our study, we suggest that the structural origin of the boson peak can be attributed to the MRO. Our persistent homology approach advances our understanding of the previously obscure relationships among boson peaks, localized modes, and MRO.

Our persistent homology-based analysis is not limited to a-Si. To further illustrate its applicability, we conducted additional simulations in which we systematically varied the tetrahedrality parameter[63–65] in the SW potential, $\lambda_{SW}$. In the main part of this study, we adopt $\lambda_{SW} = 21.0$, as proposed in the original work[50]. As shown in Fig. 9, the obtained atomic structures and PDs depend on $\lambda_{SW}$. For $\lambda_{SW} = 20.0$, the

percentages of 3-, 4-, and 5-coordinated Si atoms are 3.78%, 50.5%, and 43.6%; for $\lambda_{SW} = 21.0$, 1.79%, 83.7%, and 14.3%. For $\lambda_{SW} = 22.0$, they are 1.01%, 92.5%, and 6.33%, respectively. The increase in atoms with coordination numbers $\geq 5$ and $\leq 3$ for $\lambda_{SW} = 20.0$ corresponds to a decrease (increase) in the number of birth–death pairs in region 1 (region 2) of the PD, which indicates that the tetrahedral units are more prone to collapse, leading to a less pronounced MRO. Conversely, for $\lambda_{SW} = 22.0$, atoms with coordination numbers $\geq 5$ and $\leq 3$ decrease, suggesting the formation of more rigid tetrahedral units, which develop MRO and microcrystalline domains. Despite these structural variations, in all three cases, the local structures surrounding atoms with large nonaffine displacements consistently exhibit a hierarchical organization, wherein short-range disorder is embedded within the MRO. This hierarchical structural motif is also strongly correlated with low-energy localized vibrational modes (see the Supplementary Information). These results suggest that, despite the limitations of the simple empirical SW potential, the correlation among hierarchical structures, nonaffine displacement, and low-energy localized vibrational modes may be a general feature of tetrahedral covalent networks.

Amorphous $AX_2$ covalent solids such as a-SiO$_2$ and a-GeSe$_2$, with well-defined tetrahedral networks, are promising candidates to test this hypothesis. Binary metallic glasses, such as Pd–Si and Cu–Zr alloys, are another class of intriguing comparative targets. These materials contain polyhedral clusters as structural units, suggesting that our method is applicable. However, challenges, including determining whether first- or second-order diagrams are more suitable for capturing structural features[30,66,67] and extending persistent homology to consider multiple elemental species, must be addressed. Integrating these analyses with high-accuracy machine-learning potentials[68–70] presents a promising avenue for investigating the key correlations among hierarchical structures, MRO, nonaffine displacement, and localized vibrations across various glassy materials.

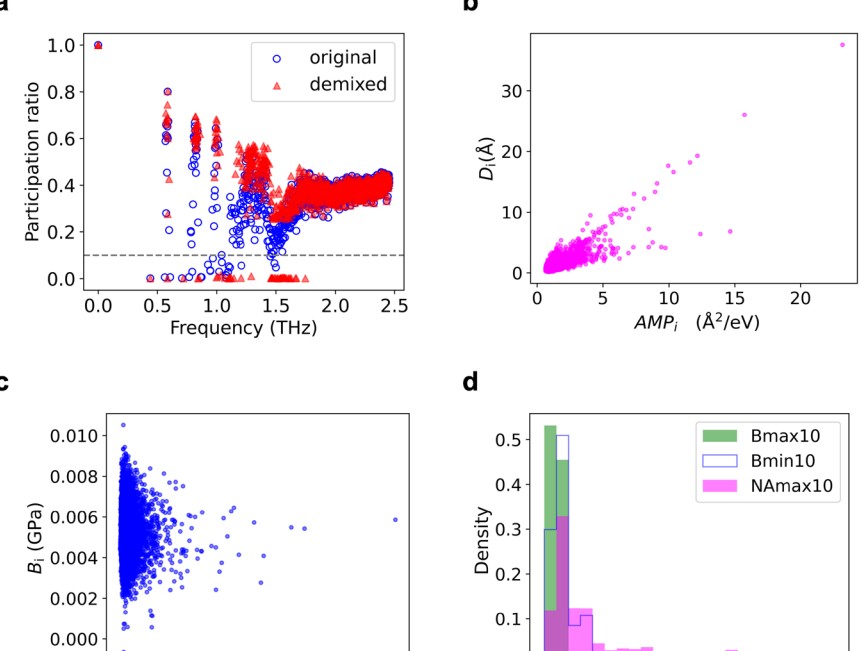

**Fig. 8 | Correlation between low-energy localized modes and elastic moduli.** **a** Calculation results of PR as a function of mode frequency. Gray dashed line indicates the threshold for identifying the low-energy localized modes. Blue open circles indicate the original calculation results of PR, and red filled triangles indicate the results obtained from the demixing procedure. **b** Nonaffine displacement ($D_i$) plotted as a function of the amplitudes of the low-energy localized modes ($AMP_i$). **c** Born term ($B_i$) plotted as a function of $AMP_i$. **d** Histograms of the $AMP_i$ values evaluated for atoms in the rings assigned to the Bmax10, Bmin10, or NAmax10 groups. Source data are provided as a Source Data files.

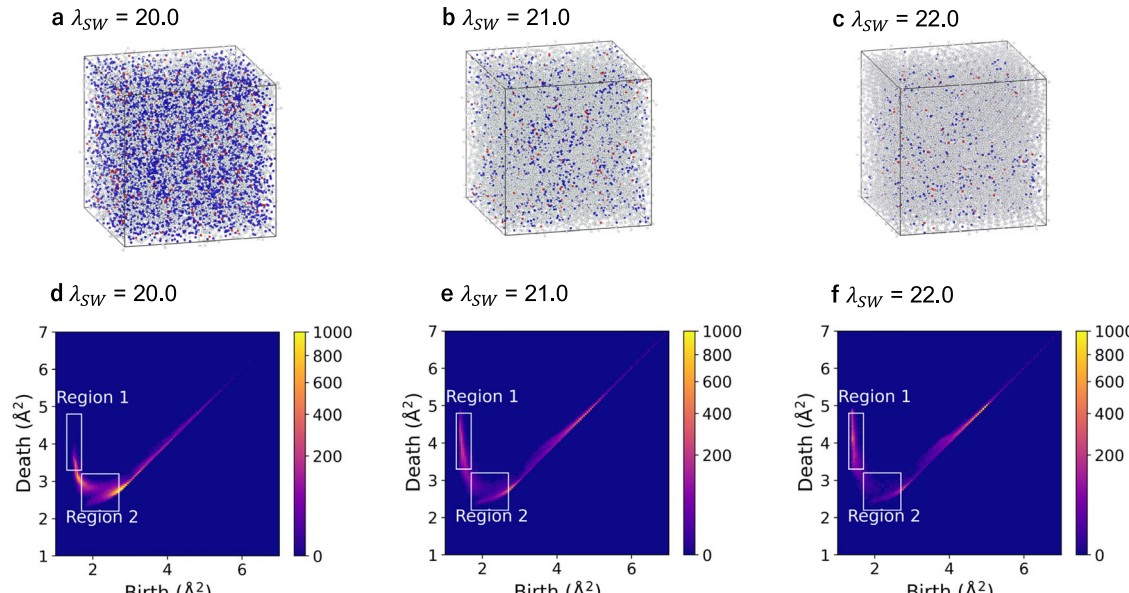

**Fig. 9 | Atomic structures and corresponding persistence diagrams of amorphous models with silicon-like bonding. a–c** Amorphous structure comprising 13,824 atoms generated with tetrahedrality parameter $\lambda_{SW}$ = 20.0, 21.0, and 22.0. White atoms represent the 4-coordinated Si atoms. Blue and red atoms represent the Si atoms with coordination numbers 5 and ≤ 3, respectively. **d–f** Persistence diagrams derived from the corresponding structures, visualized as a two-dimensional histogram.

Moreover, although this study focuses on elastic moduli, our approach with persistent homology can be extended to plastic deformation. Identifying the regions in which plastic deformation occurs is a challenging problem that has been widely debated among various researchers[21,71–83]. Several previous studies have been designed to elucidate plastic events within an energy-landscape framework. In this framework, the regions in which plastic events occur, which are often referred to as soft spots or defects, are identified in terms of low-energy localized modes[21,71–73,77,79,80], nonlinear energy excitations[74,75], instantaneous normal modes[77–79], and nonaffine displacement fields[76]. Our persistent homology approach can extract structural characteristics from regions exhibiting localized excitations and large nonaffine displacements and can further aid in identifying sites of plastic deformation. Our findings lay the groundwork for future research aimed at understanding and designing the dynamics and mechanical responses of amorphous materials based on their static structures.

## Methods
### Elastic moduli and nonaffine displacement in amorphous materials

Elastic constant $C_{\alpha\beta\kappa\chi}$ in amorphous materials can be defined by using the Born term, $C_{\alpha\beta\kappa\chi}^{\text{Born}}$, and nonaffine term, $C_{\alpha\beta\kappa\chi}^{\text{NA}}$, as[16]

$$C_{\alpha\beta\kappa\chi} = C_{\alpha\beta\kappa\chi}^{\text{Born}} - C_{\alpha\beta\kappa\chi}^{\text{NA}}, \tag{5}$$

where

$$C_{\alpha\beta\kappa\chi}^{\text{Born}} = \frac{1}{V}\frac{\partial^2 U}{\partial\eta_{\alpha\beta}\partial\eta_{\kappa\chi}}, \tag{6}$$

and

$$C_{\alpha\beta\kappa\chi}^{\text{NA}} = \sum_i\sum_{j\neq i}\frac{1}{V}\Xi_{\alpha\beta}^i\left(\mathbf{H}^{-1}\right)_{ij}\Xi_{\kappa\chi}^j. \tag{7}$$

Here, $U$ is the potential energy of the system, $\eta_{\alpha\beta}$ is the strain, $\alpha, \beta, \kappa$, and $\chi$ are the indices specifying directions $x, y$, or $z$, and $i$ and $j$ are indices specifying the atoms. Furthermore,

$$\Xi_{\alpha\beta}^i = \frac{\partial^2 U}{\partial\mathbf{r}_i\partial\eta_{\alpha\beta}}, \tag{8}$$

and

$$\mathbf{H} = \frac{\partial^2 U}{\partial\mathbf{r}_i\partial\mathbf{r}_j}, \tag{9}$$

where $\mathbf{r}_i$ is the vector indicating the position of the $i$-th atom.

Under a small strain, the derivative with respect to strain, $\eta_{\alpha\beta}$, can be obtained by applying the derivative with respect to the interatomic vector defined in the undeformed coordinate as

$$\frac{\partial U}{\partial\eta_{\alpha\beta}} = \sum_i\sum_{j\neq i}\frac{1}{2}\left(\frac{\partial U}{\partial r_{ij}^\alpha}r_{ij}^\beta + r_{ij}^\alpha\frac{\partial U}{\partial r_{ij}^\beta}\right), \tag{10}$$

where $\mathbf{r}_{ij}$ is the vector pointing from atom $i$ to atom $j$ and $\mathbf{r}_{ij} = \left(r_{ij}^x, r_{ij}^y, r_{ij}^z\right)$. According to this definition, $C_{\alpha\beta\kappa\chi}^{\text{Born}}$ is given by

$$\begin{aligned}C_{\alpha\beta\kappa\chi}^{\text{Born}} = \sum_i C_{i,\alpha\beta\kappa\chi}^{\text{Born}} = &-\frac{1}{4}\left(\delta_{\alpha\kappa}T_{\beta\chi} + \delta_{\beta\kappa}T_{\alpha\chi} + \delta_{\alpha\chi}T_{\beta\kappa} + \delta_{\beta\chi}T_{\alpha\kappa}\right)\\&+\sum_i\sum_{j\neq i}\frac{1}{4V}\left(\frac{\partial^2 U}{\partial r_{ij}^\alpha\partial r_{ij}^\kappa}r_{ij}^\beta r_{ij}^\chi + \frac{\partial^2 U}{\partial r_{ij}^\alpha\partial r_{ij}^\chi}r_{ij}^\beta r_{ij}^\kappa + \frac{\partial^2 U}{\partial r_{ij}^\beta\partial r_{ij}^\kappa}r_{ij}^\alpha r_{ij}^\chi + \frac{\partial^2 U}{\partial r_{ij}^\beta\partial r_{ij}^\chi}r_{ij}^\alpha r_{ij}^\kappa\right),\end{aligned} \tag{11}$$

where $\mathbf{T}$ is the Cauchy stress tensor, defined as

$$T_{\alpha\beta} = \frac{1}{V}\sum_i\sum_{j\neq i}\frac{\partial U}{\partial r_{ij}^\alpha}r_{ij}^\beta. \tag{12}$$

As described above, the Born term can be expressed as the sum of the contributions from each atom ($C_{i,\alpha\beta\kappa\chi}^{\text{Born}}$). However, it is difficult to decompose nonaffine terms in this manner. This is because the definition of the nonaffine term includes $\mathbf{H}^{-1}$, the inverse of the Hessian matrix. The eigenvalues of the Hessian matrix correspond to the

squares of the vibrational mode energies. In solids, Goldstone modes with zero energy appear because of the translational symmetry. This makes the Hessian matrix singular and its inverse ill-defined. Therefore, in the calculation of $C^{NA}_{\alpha\beta\kappa\chi}$, the Hessian matrix with one row and column removed to eliminate the excess degrees of freedom due to translational symmetry is used to evaluate the inverse[18]. The value of $C^{NA}_{\alpha\beta\kappa\chi}$ obtained is invariant with respect to the choice of fixed atom. However, this treatment is inappropriate when the nonaffine term is decomposed into contributions from each atom.

Although it is difficult to decompose the nonaffine term, such an analysis is possible for nonaffine displacements (Eq. (2)). The nonaffine displacement is defined as

$$\left(\mathbf{D}^{NA}_{i,\alpha\beta}\right)_k = \frac{\partial r_{ik}}{\partial \eta_{\alpha\beta}} = \sum_{j\neq i}\sum_{l=x,y,z}\left(\mathbf{H}^{-1}\right)_{ikjl}\frac{\partial^2 U}{\partial r_{jl}\partial \eta_{\alpha\beta}}. \tag{13}$$

### Molecular dynamics simulation to generate amorphous structures

The structure of a-Si was created by using LAMMPS to implement a classical molecular dynamics simulation[84]. All the molecular dynamics simulations were performed by using an NVT ensemble. The system temperature was controlled by applying a Nosé–Hoover thermostat[85,86] with a 1-fs time step. The interactions between Si atoms can be described by applying the SW potential[50]. Our a-Si system model contained $N = 13{,}824$ atoms in a cubic unit cell. The length of each unit cell, $L_x = L_y = L_z = L$, was approximately 65 Å. The resulting mass density of the system was 2.35 g/cm³. The method for creating the amorphous structure followed the procedures described in the literature[43,87]. First, crystalline Si was heated to 3510 K for 500 ps to melt it into a liquid state. After the system was equilibrated at 3510 K for 500 ps, it was quenched to 10 K at a cooling rate of $10^{11}$ K/s. Then, after equilibration at 10 K for 500 ps, the system was annealed at 100 K for 500 ps. Subsequently, all atomic velocities were set to zero and structural relaxation was performed. The resulting inherent structure was used as an amorphous model and visualized by VESTA[88]. The shear modulus of the amorphous structure was 33.6 GPa (Born term: 73.7 GPa; nonaffine term: 40.1 GPa), which is consistent with previous reports[43].

For statistical analysis, nine additional independent samples with different initial velocities were created in addition to the main sample analyzed in this study. The average shear modulus of the structures of the 10 samples was 33.2 GPa, with a standard deviation of 0.36 GPa. The averages (standard deviations) of the Born term and nonaffine term were 73.5 GPa (0.34 GPa) and 40.3 GPa (0.18 GPa), respectively.

### Persistent homology

HomCloud code[89,90] was used to apply the persistent homology method, focusing on the first homology. According to convention, we applied the squared values of the radii of the spheres used in the filtration procedure as the birth and death radii (unit: Å²). We identified the ring structure corresponding to each birth–death pair via the inverse analysis. In some cases, there are possible homologically equivariant candidate structures. To find the optimal structure from candidates, we applied the stable volume[53] and volume-optimal cycle[52] methods, both of which minimize the volume enclosed by the determined structure.

The stable volume method enhances robustness to noise by tightening the volume based on noise strength, providing more interpretable results. However, it cannot be applied to pairs close to the diagonal of the persistence diagram. For such cases, the volume-optimal cycle method was used instead.

For pairs for which death radius − birth radius > 0.1, the optimal structure was determined from the stable volume by setting the noise level to 0.001. For other cycles near the diagonal, the optimal structure was determined by using the volume-optimal cycle.

## Data availability

The source data corresponding to all scatter plots and line graphs in the main text and Supplementary Information are provided in the Source Data file. Additional data, including atomic coordinates, analysis scripts, and simulation outputs, have been deposited in the Zenodo repository under the accession code https://zenodo.org/records/15959771. Source data are provided with this paper.

## Code availability

The code used in this study is available at GitHub and has been archived in Zenodo for reproducibility[91]. The archived version is accessible at https://doi.org/10.5281/zenodo.15959267.

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

## Acknowledgements
The authors thank Tetsuya Morishita for valuable comments on the MD simulations and structural analysis. This study was supported by JST PRESTO Grant Number JPMJPR2198 (E.M.), JPMJFR236Q (E.M.), MEXT KAKENHI 21H01816 (E.M.), 23H04470 (E.M.), 22K03543 (H.M.), 23H04495 (H.M.), 19KK0068 (I.O.), 20H05884 (I.O.), and 22H05106 (I.O.), and a grant from the Inamori Foundation (E.M.).

## Author contributions
E.M. conceived the research and performed the simulations in the main text. T.N. and I.O. assisted in the persistent homology analysis. H.M. performed the mode-demixing procedure and assisted in the numerical analysis of elastic moduli in a-Si. All authors contributed to the discussion, data interpretation, and writing of the manuscript.

## Competing interests
The authors declare no competing interests.
