## [Transparent Peer Review file · Nature Communications]

Persistent homology elucidates hierarchical structures responsible for mechanical properties in covalent amorphous solids

Corresponding Author: Professor Emi Minamitani

Version 0:

Reviewer comments:

Reviewer #1

(Remarks to the Author)

The authors have extended the capabilities of persistent homology analysis to understand the relationship between structure and mechanical properties. While their approach is interesting, we do not recommend their publication in Nature Communications because the content is too specialized and I do not see significant benefit for the general amorphous researchers.

1. I am wondering that the reason why the authors chose amorphous silicon. As described in ref. 19., medium range order has been discussed for AX₂ type amorphous materials and there is little discussion of amorphous silicon and the description is very unclear.

2. The descriptions of SRO and MRO for amorphous silicon are very unclear and difficult to understand for the broad readership of Nature Communications. It is necessary to describe with length scale associated with coordination and connectivity since the authors have an atomistic configuration of amorphous silicon. I recommend the authors to read the following article.

Philip S Salmon and Anita Zeidler, J. Stat. Mech. (2019) 114006.

3. There is no general introduction to amorphous materials and amorphous silicon. This is not helpful to the broader leadership of Nature Communications.

4. It is not easy to understand what Fig. 5 means, particularly Fig. 5c. A comparison with experimental data might be helpful. I recommend the authors to read the following article.

Khalid Laaziri et al., Phys. Rev. Lett. 82 (1999) 3460.

5. I cannot understand what "pairwise distances within ring structures" means.

Reviewer #2

(Remarks to the Author)

The present manuscript of Minamitani et al. reports on the coupling between atomic-scale structures identified using so-called persistent homology and the intrinsic local elastic properties of amorphous Si. The authors generate a structure of amorphous silicon using molecular dynamics simulations to followingly analysis it by structural, mechanical, and persistent homology analysis. This brings them to identify a relation between atoms of large non-affine displacement which are found to have local structures of more vertices compared to atoms showcasing small non-affine displacements.

Generally, I think the work is of great interest to the community and has a significant potential in how to structurally identify regions of mechanical instability. However, I think the presented works lacks generality amongst more disordered/amorphous material families, fails to address important parts of the literature, and could greatly benefit from a

more detailed analysis of the already presented data. Please see my detailed response below.

- Page 2: Stating that “The correlation between structure and properties of amorphous materials has long been a mystery in materials science, even for the fundamental mechanical properties” is to me a somewhat bold statement. Structure-property building in amorphous materials science has been ongoing for decades. This is evident in a number of cases with structure being correlated to a number of both thermal and mechanical properties, e.g. using coordination numbers or topological constraints as structural metrics for model-building. A few useful examples might be <https://doi.org/10.1016/j.commat.2018.12.004> and <https://doi.org/10.1016/j.jnoncrysol.2013.07.028>
- Page 2, second paragraph: I think the authors overlook the recently published metric of softness (<https://doi.org/10.1126/science.aai8830>) – this metric is parameterized to quantify atoms as either soft or hard based on their local environment. While it is not a direct replacement of the suggested persistent homology method, the authors may consider discussing its relevance or even using it as a source of comparison to e.g. their vertices counting.
- Page 3 paragraph 1, please provide some examples of “conventional structural analysis methods”
- Page 3-5. I want to thank the authors for including a rather nice introduction to the concept of persistent homology. I think such paragraph is worthwhile for most readers as persistent homology is not yet well-established as a method in the materials science community. However, I think the paragraph also deserves to mention some of the drawbacks of the persistent homology method. I suggest the authors to at least discuss this.
- Figure 1. Subfigure e is not referenced in the caption. Please provide a better and more elaborate caption text.
- Figures in general. Please provide larger text in the labels of the figures. Furthermore, why is there a difference in the amount of significant figures in the figure axes (even within the same subfigure)? For all figures please provide more descriptions of abbreviated variables in the captions – ideally to an extent where the figures can be understood independently from the main text.
- Figure 2: I suggest the authors to color code the atoms in Figure 2b according to their Born and non-affine displacement. Perhaps as subfigures c and d?
- Page 6 bottom: The authors choose to focus on only 10 atoms in each group. This is less than 0.1% of the atoms in the sample. While I understand it is indeed interesting to focus on the atoms of very high or low B_i and D_i , the authors should at least try to broaden their analysis to include more atoms, e.g. logarithmically by testing with 10, 20, 50, and 100 atoms. Simulations are somewhat prone to always have some extreme atoms which might not be very representative.
- Page 7, top: Please elaborate further on what you mean by “inverse analysis”.
- Page 7, top: In addition to the above comment, I assume this practically means going from an identified loop back to an atomic representation. As I guess the authors are aware this comes with a number of difficulties (or at least, active choices). In the methods you state that you employ the “stable volume” and “volume optimal cycle” in the inverse analysis. Please elaborate further on this in the main text in terms of what this specifically means for the identified loops and why you choose these over alternatives (shortest path, least atoms, etc. etc.).
- Page 7, top: You state that: “...that included B_{max10} , B_{min10} , and N_{Amax10} ”. This needs elaboration. Does all 10 atoms in each group has to be in the loops for it to be analyzed? Just one of the atoms or maybe some?
- Figure 3: I suggest making the points (or at least legends) larger and clearer. Furthermore, I wonder why the three datasets are all shown twice instead of just plotting them all in a single plot? I believe this division into three subfigures (3b-d) makes comparison harder.
- Figure 3 and related discussion: Often, persistence diagrams are very qualitative and difficult to interpret. I suggest the authors to use more quantitative measures for drawing their conclusions, e.g. the recently developed accumulated persistence function, see.: <https://doi.org/10.1080/10618600.2019.1573686>
- Figure 5: I suggest the authors also to compare between the graphs vertically. While some differences between graphs are clear (e.g. in Fig. 5a) others are more subtle (e.g. 5c).
- Page 9, second paragraph: The authors identify region 1 and region 2 to have very high and low vertex counts, respectively. Given previous literature on oxides (Refs. 20 and 26 in the paper), this is hardly any surprise as longer rings with more atoms will inherently be found in low-birth, high-death regions (such as region 1). Please discuss further and provide more elaborate referencing to existing literature.
- Page 12, bottom: I suggest the authors to introduce and discuss the use of Eq. 12 in the main text and not in the Methods section
- Page 13, top: The authors state “... for 1000 low-energy vibrational modes”. Do you mean the 1000 modes of lowest eigenfrequencies? What frequency range does this correspond to?

- Page 13: A minor thing, but IPR is usually coined as “inverse participation ratio” (not inversion).
- Page 13: I suggest the authors to provide the fundamental limits of the IPR (1 and N) as well as provide a plot of the participation ratio (that is, the inverse of IPR) – at least in the SI. Usually, highly localized modes are given as modes with a $PR < 0.15$ but it is currently very difficult to judge exactly how localized the studied modes in fact are.
- Eq. 2 please provide a description of the lambda variable.
- Page 14: I suggest to significantly deepen the analysis discussion on the connection between the identified structural features and vibrational properties. Currently it seems somewhat vague and superficial. One could test e.g. the correlation between B_i , D_i , and IPR and possibly show in the SI. For more, the very low-frequency modes are responsible for a large amount of literature which seems to be somewhat overlooked, e.g. in relation to the so-called boson peak, see e.g. <https://doi.org/10.1103/PhysRevLett.117.035501>.
- Page 18: You write: “, which is consistent with previous reports.” Please provide suitable references for this statement.
- Generally, I think the biggest caveat of the study comes to how only amorphous Si is tested. While I acknowledge that amorphous Si is a very thoroughly tested and good reference material it is also one of the absolute simplest disordered materials both structurally and dynamically. As such, while the present study serves somewhat as a proof of concept, the lack of generalization to other amorphous systems (oxides, chalcogenides, metals, halides, nitrides, all possibly with multiple components) significantly dampens the possible applicability of the suggested method.

Reviewer #3

(Remarks to the Author)

This paper studies local structures induced by Born terms and large non-affine terms by means of persistent homology analysis, and furthermore, tries to find a relationship with low-energy localized vibrational excitations. The analysis tools they present look powerful to identify subtle but significant atomic structures to distinguish those two terms. The computational code of the persistent homology analysis (HomCloud) is open to the public, so the reader can easily follow their analysis to other materials. I believe that the proposed analysis and results will be of interest to a broad audience of the journal. Therefore, I basically would like to accept this paper for publication, under the condition that the authors take care of the following concerns and suggestions as detailed below.

1. I think that the result between line-191 and line-201 is significantly important for the discussions afterwards, but the discussion here seems to be not quantitative. I strongly recommend the authors to modify this part so that the conclusions will be derived in more quantitative way. One idea may be to use some distance on persistence diagrams to measure the similarity/difference.
2. The discussion from line-236 to line-250 should be revised more logically. The point I am not satisfied is that, since the definition of SRO and MRO in this paper is ambiguous (at least for me), the assignment of SRO and MRO to the local structures looks ambiguous and less convincing. I would like to understand this part more by adding further logical and quantitative explanations.
3. Unfortunately, the quality of English seems to be not satisfactory. I will not raise all those parts here, but I recommend the authors to seriously recheck English in the manuscript.

Reviewer #4

(Remarks to the Author)

The manuscript presents molecular dynamics simulations of a 13,824-atom structure of amorphous silicon, described with the Stillinger-Weber potential [Phys. Rev. B Condens. Matter 31, 5262–5271 (1985)]. Molecular dynamics is used to compute the affine and non-affine components of the atomic displacements from equilibrium. Then, the authors look for a correlation between persistent homology analyses for atoms having particularly large or small affine and non-affine displacements.

The manuscript further attempts to connect hierarchical structures identified by the persistent homology analysis with low-energy localized vibrational excitations. To the best of my understanding, this connection is the novel part of the work, but I am not convinced this meets the standards for publication in Nature Communications for the following reasons:

1) The main messages of the manuscript are not clearly explained. The manuscript is written in a style suitable for a highly specialized journal, not a high-impact journal such as Nature Communications. To the best of my understanding, there are two main messages in this work:

A) The existence of a relation between low-energy localized vibrational excitations and hierarchical structures emerging from the persistent homology analysis;

B) The connection between atomic displacements, structures emerging from the persistent homology analysis, and elastic moduli.

These points are not clearly explained, and only an expert reader familiar with the field can extrapolate this after reading the manuscript multiple times. For example, all the details on the elastic moduli are reported in the methods, and there is no figure to highlight the connection between structures emerging from the persistent homology analysis and elastic moduli.

2) One of the most important claims made by the authors (1A) involves the localization of vibrational modes in disordered systems. As discussed in [H. R. Schober and C. Oligschleger PRB 53 1996 <https://link.aps.org/doi/10.1103/PhysRevB.53.11469>], and in more recent articles that cite this work, the localization properties of low-frequency vibrational modes in disordered systems are subject to spurious, non-physical effects caused by periodic boundary conditions. The article above discusses a procedure to correct for these spurious effects. The authors simulate a structure containing 13,824 atoms using molecular dynamics in periodic boundary conditions. They do not provide evidence that their results (e.g. Fig. 8) are physically meaningful and not affected by these spurious finite-size effects.

3) The computational methodology employed is outdated, based on the Stillinger-Weber potential [Phys. Rev. B Condens. Matter 31, 5262–5271 (1985)] developed 39 years ago. The authors could have done the same analysis using state-of-the-art and quantum-accurate machine-learning potential. The persistent homology analysis based on the HomCloud software is useful but not novel. The novelty of the present work stands in the relation between structural motifs emerging from the persistent homology analysis and elastic moduli, but as mentioned at point 1 this is not clearly explained.

Minor comments:

4) Could the author clarify and expand the following sentence?

Because homology groups are defined as the quotient group of cycles by boundaries, the transformation from cycles to boundaries is fundamentally important.

In particular, what do they mean by "the transformation from cycles to boundaries is fundamentally important"?

5) Which message are the authors trying to convey with Fig. 2b? If the message is that the structure is disordered, this is not clearly conveyed in Fig. 2b.

6) The authors mention multiple times "scale of short-range order (SRO)" or "scale of medium-range order (MRO)", but do not clearly quantify such scale. Could the authors elaborate that more clearly, and highlight (ideally in a figure, e.g., in an extension of Fig. 1) the connection between the persistence diagram and SRO/MRO scale.

Version 1:

Reviewer comments:

Reviewer #1

(Remarks to the Author)

Reviewer #2

(Remarks to the Author)

In general, the authors nicely addressed my queries and I believe the revision has significantly improved the manuscript.

One thing I noticed going through the revision was the definition of the localised modes in the new Figure S8. The authors state in the text how they use a definition of localization of $PR < 0.1$ (which I agree is reasonable), but in the figure they point out modes with PR below ~ 0.01 . The authors may try to explain/correct this discrepancy.

One last comment: My main concern still beholds to my previous very last comment on how only a-Si is tested in the presented work. While I do acknowledge that the extension to other systems come with a range of new problems (especially that of varying initial radii arising from multiple atoms), these problems have been somewhat nicely addressed e.g. for simple oxides such as SiO₂ (e.g. in refs. 30, 32, and 40 in the references of the revised manuscript) and that such computation is not wildly demanding to perform. As such, I maintain that the impact of the work would be somewhat improved by testing/showcasing the applicability of the method beyond that of the very simplest case of a-Si.

Reviewer #3

(Remarks to the Author)

I confirmed that the authors appropriately revised the manuscript in view of the comments I raised in the report. Now I am confident that this paper is well qualified to be published from the journal.

Reviewer #4

(Remarks to the Author)

Unfortunately, the concerns I raised in the last review report have not been adequately addressed. As a result, I have maintained my original review opinion and, therefore, can not recommend the manuscript for publication.

Key concerns:

1. A quantitative discussion on finite-size effects is missing.

The primary claim of the manuscript is supported by the term “should” without providing any quantitative evidence. For example, the authors state:

“We describe the finite system-size effects for results of the localized modes, which arise from hybridizations with spatially extended vibrational modes.”

This statement is unclear. How exactly do the authors describe “the finite system-size effects for results of the localized modes”? The manuscript lacks any quantitative discussion of how affine and non-affine displacements are influenced by the system size. There is ongoing debate in the literature about the persistence or disappearance of localized modes as system size increases, which the authors have completely neglected.

The authors further state:

“To eliminate these effects, we need to use literature methods^{51,52}. However, this hybridization primarily affects the vibration in the region far away from the core region where the amplitude of localized mode is large^{53,54}.”

This sentence is both unclear and misleading. Among the references cited in the article, several studies show how hybridization affects low-energy modes that have frequency similar to the red and blue points in Fig S8d.

The statement:

The analysis in Figure 8 focuses on the localized core region with large AMP_i in relation to the Born term and nonaffine displacements, which should remain unchanged even if we consider the hybridization.

This is a far-reaching claim that is not quantitatively justified, and based on Fig 3 and Fig 4 in <https://link.aps.org/doi/10.1103/PhysRevB.53.11469>, probably incorrect.

The authors conclude:

“Therefore, we conclude that our findings are not influenced by finite system-size effects.”

This conclusion is not substantiated by any quantitative evidence, and based on the arguments above, I am skeptical about its correctness.

2. Accuracy of the empirical SW potential

The authors attempt to justify the accuracy of the empirical potential by presenting the radial distribution function (RDF) in Fig. R2. While the Stillinger-Weber (SW) potential may describe structural properties well, there is no evidence that it accurately captures the subtle differences between affine and non-affine vibrations. Quantitatively describing such subtle effects likely requires state-of-the-art simulation methods.

For instance, defining bonding in amorphous silicon is inherently challenging, as discussed from first principles in Solid State Communications, Vol. 107, No. 1, pp. 7-11, 1998. Accurately describing the coordination properties of amorphous silicon is beyond the capabilities of the SW potential and may only be feasible using modern machine-learning methods.

3. General lack of attention to detail

The manuscript demonstrates a lack of attention to critical details, which compromises its suitability for publication in Nature Communications. Examples include:

- Equation 4 contains dimensionless eigenvector components alongside the reciprocal eigenvalue of the Hessian matrix (λ_p^{-1}). Since λ_p represents the square of vibrational mode energy, it has units. Why is AMP_i reported in plots without units?

I also note that a previous referee comment (2-21) explicitly asked for a description of the λ variable, yet the authors have still omitted the units.

- The plot 8a is presented in a way that makes it difficult to understand it. The mode index does not have physical meaning, as it depends on the model's size. The authors should consider using other physical properties for the x axis of this plot, such as the frequency of the modes (as they did e.g. in Fig R11b).

4. Flow and clarity

The revisions have only marginally improved readability, the flow of the article remains challenging to follow.

Version 2:

Reviewer comments:

Reviewer #1

(Remarks to the Author)

The authors have improved the ms. again, but I still have three issues.

- 1) I think the agreement between MD simulation and XRD data in Q space is good. But the authors did not properly Fourier transform experimental data. So I recommend the authors to omit the comparison in $J(r)$.
- 2) I understand that the Si coordination number has a distribution. I ask the authors to show the distribution of coordination number, which is an important point.
- 3) I do not understand what short-range disorder means. This needs to be clarified.

Reviewer #2

(Remarks to the Author)

With the new revisions made by the authors, I think the manuscript has become more rigorous, especially with the inclusion of more detailed testing using varying Stillinger-Weber parameters. Although I would have enjoyed seeing extension to further systems, I acknowledge the difficulty of this and that the present changes satisfies my previous queries of the manuscript.

Reviewer #4

(Remarks to the Author)

The authors have made an effort to address my previous comments. However, several important aspects of the analysis still require clarification and further development.

The statement in Fig. 8—“The correlation between AMP_i and nonaffine displacement remains unchanged [between original and demixed modes]”—remains unclear and requires a more careful analysis. In particular, it is surprising that the authors do not show how demixing influences the D_i vs. AMP_i and B_i vs. AMP_i relationships. It would strengthen the analysis to include both the original and demixed modes in Fig. R5b, to clarify whether and how the correlation (or lack thereof) is impacted by the demixing.

In the previous version, the localization threshold was somewhat arbitrary, since without demixing the IPR distribution does not show a clear separation between localized and delocalized modes.

After the demixing, the identification of localized low-energy modes appears more well-defined, yet this critical improvement is not clearly explained in the revised manuscript. Fig. 8a shows significant hybridization in the blue points, and based on the Schober-demixing references the authors cited, I suspect only the red points are physically meaningful. This important aspect deserves further discussion and proper explanation. I would suggest checking that the demixed modes do not depend on the model's size, while the original modes depend on the system size. This standard size analysis is normally performed in all studies of amorphous materials, and a proper investigation of size effects was suggested in my previous report.

If this is correct, then it would imply that demixing is critical to unambiguously define the low-energy localized vibrational modes that display correlation with the hierarchical structures and nonaffine displacements.

With regard to methodology, the justification for using the Stillinger-Weber (SW) potential relies on references that are now at least three years old, and likely among the last to adopt this approach. Since the publication of Deringer et al., Nature 589, 59 (2021), there has been a clear shift in the community away from empirical potentials for modeling amorphous silicon.

Finally, I encourage the authors to moderate the scope of their conclusions. The statement that “the results demonstrate that the correlation between the hierarchical structures, nonaffine displacement, and low-energy localized vibrational modes is a general feature of tetrahedral covalent networks” is far-reaching and might be an artifact of the simplified form of the SW potential. In particular, coordination defects are much more prevalent in a-Si than in a-SiO₂, and the manuscript does not discuss whether the empirical modified SW potential can capture these differences.

In summary, while the manuscript presents interesting results, it would benefit from clearer presentation and interpretation of its findings. I strongly recommend that the authors avoid speculative conclusions tied to the SW potential. With these improvements, the work could be suitable for publication, though I remain unsure whether it will attract wide attention in a field that is rapidly moving beyond empirical potentials.

Version 3:

Reviewer comments:

Reviewer #1

(Remarks to the Author)

The authors have tried to improve the ms. with considering the reviewer's comments.

I still have several issues.

1. About Fourier transform, I understand the procedure of PRB paper. I attach the data w/o Lorch function. Please compare with my attached numerical data. Experimental data in Fig. S1 looks strange (very noisy) to me. But I accept the final decision of the authors for Fig. S1.

2. I think $\lambda_{SW}=21.0$ is the best MD model. If this is correct, I am a bit surprised about the low fraction of fourfold silicon atoms. This needs to be discussed.

3. If the authors discuss short-range disorder, it is necessary to show the bond angle distribution.

Reviewer #4

(Remarks to the Author)

The authors have made an effort to address my comments. I still think their answer contains some unclear aspects.

In particular, in their response they write:

"In Figure R3, original eigenvectors exhibit strong size dependence across the entire frequency range, except for a few low-frequency localized modes."

and later:

"In AMPi calculation (Response to Comment 1), AMPi values are dominated by low-energy localized modes, so the demixing procedure had minimal impact on conclusions."

I do not understand how they can claim that "the demixing procedure had minimal impact on conclusions."

Figure R3 shows a clear gap between localized and delocalized modes only when demixing is used. Without demixing, the definition of localized modes would have been arbitrary, and they should note that the threshold 0.1 shown in their Figure R2 is arbitrary.

After the authors elaborate on this point in the text, I believe the article will be technically sound. I remain unsure whether the work will attract broad attention, given the field's ongoing shift beyond empirical potentials, but I think it can be published, and time will tell.

Version 4:

Reviewer comments:

Reviewer #1

(Remarks to the Author)

The authors conducted thorough research into previous studies and eventually discovered the modified SW results. I therefore recommend that the authors perform MD simulations and repeat the author's PD analysis. Otherwise, I strongly recommend that the authors move Fig. S14 into the main text with the coordination number distribution of the modified SW model. In this case, the authors should emphasise that the objective of the study is not to reproduce the atomic structure or coordination number distribution of a-Si. Perhaps, it would be better to refer J. Non-Cryst. Solids 282, 248–255 (2001) in the introduction and mention the objective mentioned above to prevent misleading to the broad audience of Nature Communications. In this case, unfortunately, the comparison with diffraction data and the visualisation of the coordination number distribution in atomistic models would not make sense.

Reviewer #4

(Remarks to the Author)

The authors addressed my comments.

I think the work might be suitable for publication, though I remain unsure whether it will attract wide attention in a field that is rapidly moving beyond empirical potentials.

Version 5:

Reviewer comments:

Reviewer #1

(Remarks to the Author)

The authors replied to all of my queries. I have appreciated the authors' responses for a long time. It is a really nice paper with PD.

The author has made significant revisions to the original text. This is highly commendable. However, there are serious problems with the structural model created by the MD simulations that warrant reconsideration. I am also afraid that the structural information obtained from the PD analysis is too specialized and that this paper would be better suited for publication in a more specialized journal. Therefore, we cannot accept this paper in its present form.

1. The agreement between the experimental $S(Q)$ and $g(r)$ and computed ones in comparison with data reported by V. L. Derin *et al*, *Nature*, **589**, 59 (2021)(GAP-MD) and in "Computer Simulations of Glasses: Methodologies and Applications", Ed. by J. Du and A. N. Cormack, Wiley-American Ceramic Society, Hoboken (2022) pp. 60 (MD-RMC). I am really afraid that we cannot see good agreement even when the Q_{\max} value is extended to 20 \AA^{-1} .
2. I feel that the experimental $g(r)$ calculated by the authors using the Fourier transform of the $S(Q)$ data is very different from the original data reported in PRL. The Fourier transformed data by me is also attached below.

3. The fact that the Si atoms are not exactly fourfold in the revised ms. is inconsistent with previous reports and is the most important issue for me; this point needs to be discussed substantially before the PD analysis.
4. I recommend the authors to consult an expert in MD simulation, amorphous diffraction, and amorphous structure.

Point-by-point response to the Reviewer #1:

[Comment 1-1]

The authors have extended the capabilities of persistent homology analysis to understand the relationship between structure and mechanical properties. While their approach is interesting, we do not recommend their publication in Nature Communications because the content is too specialized and I do not see significant benefit for the general amorphous researchers.

[Our reply 1-1]

We appreciate the reviewer's time and effort in evaluating our manuscript. We believe that the previous version did not effectively describe the general interest and significance of our work. In this revised edition, we have implemented substantial changes, especially in the introduction, to better convey the general interest and significance of our work to a wider audience in Nature Communications. Below, we will address the issues and concerns raised by the reviewer. We hope the reviewer finds the revised version more compelling and aligned with the standards of Nature Communications.

[Comment 1-2]

1. I am wondering that the reason why the authors chose amorphous silicon. As described in ref. 19., medium range order has been discussed for AX₂ type amorphous materials and there is little discussion of amorphous silicon and the description is very unclear.

[Our reply 1-2]

We thank the reviewer for this insightful comment, which provides an opportunity to clarify the aim and novelty of our manuscript. In the revised manuscript, we have elaborated on the significance of amorphous silicon (a-Si) and its medium-range order (MRO) characteristics in the introduction.

In this study, we introduce a novel methodology based on persistent homology to elucidate the correlations between structural features and mechanical responses in covalent amorphous solids. To demonstrate the validity of our approach, we selected a-Si as a test model system for two main reasons. First, a-Si is a single-element system, which eliminates computational ambiguities such as the choice of the initial radius in persistent homology analysis. We also discuss the potential extension of our methodology to multi-element systems in the revised manuscript. Second, a-Si is an extensively studied material in both experimental and computational research and plays a critical role in electronic applications. From both fundamental and practical perspectives, a-Si represents an important amorphous material.

Additionally, as the reviewer correctly pointed out, Ref. 19 in the original manuscript primarily discusses medium-range order (MRO) in AX_2 -type amorphous materials. MRO typically refers to correlations between atomic positions within the range of 5–20 Å, which are absent in fully random structures. For covalent amorphous solids, MRO can be described by the connectivity of the polyhedral units that constitute the atomic network. The smallest scale of MRO is determined by whether adjacent polyhedra share a corner, edge, or face. Larger-scale MRO is characterized by the connections of three or more polyhedra. Since a-Si also forms a tetrahedral network structure, this definition of MRO is directly applicable.

We therefore believe that a-Si serves as the ideal system for exploring the relationship between structural features, such as MRO, and mechanical responses through persistent homology. We hope this clarification resolves the reviewer’s concerns and highlights the broader relevance of our approach.

[Comment 1-3]

2. The descriptions of SRO and MRO for amorphous silicon are very unclear and difficult to understand for the broad readership of Nature Communications. It is necessary to describe with length scale associated with coordination and connectivity since the authors have an atomistic configuration of amorphous silicon. I recommend the authors to read the following article.

Philip S Salmon and Anita Zeidler, J. Stat. Mech. (2019) 114006.

[Our reply 1-3]

We thank the reviewer for their valuable feedback and for recommending the reference paper. In response, we have significantly expanded the introduction, providing a detailed explanation of short-range order (SRO) and medium-range order (MRO) in a-Si. To better illustrate MRO in the a-Si system, we have added Fig. 1b in the revised manuscript, presented here as Figure R1. We believe these revisions make our paper more accessible to the broad readership of Nature Communications.

Figure R1: Visualization of a typical covalent amorphous network structure, using a-Si as an example. In this structure, SiSi_4 tetrahedral units share corners, and MRO is defined by the angles between these tetrahedra. When analyzing the persistence diagram (PD) for such a structure, we frequently observe parent cycles and their associated child cycles (refer to the manuscript for definitions of parent and child cycles). The blue and red polygons provide an example of a parent-child relationship. The red triangle, representing a child cycle, involves only two tetrahedra and reflects the smallest scale of MRO. In contrast, the blue hexagon, representing a parent cycle, involves three tetrahedra and corresponds to a larger scale of MRO. This demonstrates how the hierarchical structure in the PD captures the multi-scale nature of MRO.

In addition, following the reviewer's feedback and the recommended reference (Philip S Salmon and Anita Zeidler, *J. Stat. Mech.* (2019) 114006), we performed additional structural analyses of a-Si, including calculations of the structure factor $S(k)$ and the radial distribution function (RDF) for comparison with experimental results. The results are shown in Figure R2 below, also available in the supplementary materials (Figure S1). The principal peak at low k-values in $S(k)$ for a-Si reflects the height of the SiSi_4 tetrahedral structure, as described in *J. Stat. Mech.* (2019) 114006. The simulation results for $S(k)$ and RDF agree well with experimental data. In particular, the peak positions are quantitatively consistent between the simulation and experimental data. This result indicates that the bonds and network structure in the present numerical system accurately represent the real a-Si structure. In the revised manuscript, we have added these discussions and explanations on the structural properties of a-Si in the supplementary materials.

Figure R2: Results of structural analyses for a-Si model. **a** Structure factor $S(k)$. The blue solid line shows the simulation result, and open circles denote the experimental data. The insets show the SiSi_4 tetrahedral unit in the a-Si structure. The position of the principal peak

at low k -values is determined by the height of the tetrahedral unit. **b** Radial distribution function $g(r)$. The black solid (red dotted) line is the simulation (experimental) result. The experimental results in **a** and **b** taken from Ref. 1 and 2 in the supplementary materials.

[Comment 1-4]

3. There is no general introduction to amorphous materials and amorphous silicon. This is not helpful to the broader leadership of Nature Communications.

[Our reply 1-4]

We thank the reviewer for this useful feedback, as it helps us enhance the readability of our manuscript. We now acknowledge the shortcomings in our introduction. We believe that this issue stemmed from the interdisciplinary nature of the paper. Since it integrates aspects of the physics of amorphous solids, material science of a-Si, and the topological data analysis using persistent homology, it requires diverse knowledge, which may have contributed to its lack of accessibility. In response to the reviewer's feedback, we have significantly expanded the Introduction and Discussion sections to include clear and detailed explanations for each aspect. Specifically,

1. We added a detailed explanation of the concept of elasticity in amorphous materials, carefully tailored for general readers. In particular, we clarified the concept of nonaffine displacements.
2. We provided a clear explanation of the vibrational modes of glasses, focusing specifically on localized vibrational modes and the boson peak phenomenon.
3. We included introductory explanations of covalent amorphous solids, a-Si, and its MRO. Additionally, we have provided a detailed explanation of why the present work focuses on a-Si and how the findings can be extended to other amorphous materials.
4. We clearly explained our main findings. In particular, we discussed how nonaffine displacements and localized vibrations correlate with the MRO, emphasizing the contributions of this study. Additionally, we included figures to illustrate the key concepts in this study (Fig. 1a and b).

[Comment 1-5]

4. It is not easy to understand what Fig. 5 means, particularly Fig. 5c. A comparison with experimental data might be helpful. I recommend the authors to read the following article.

Khalid Laaziri et al., Phys. Rev. Lett. 82 (1999) 3460.

[Our reply 1-5]

We appreciate the reviewer's feedback, which is helpful for us in refining the manuscript. In the revised version, we have refined the description of Figure 5 and expanded its caption to provide a more detailed explanation of the presented analysis.

Figure 5 presents the results of the analysis of ring structures identified through persistent homology. The analysis was carried out for three groups: rings including atoms with large nonaffine deformations ($N_{\text{max}10}$), those with large Born terms ($B_{\text{max}10}$), and those with small Born terms ($B_{\text{min}10}$). Specifically, Figure 5a shows the distribution of vertex counts for ring structures in each group. Figure 5b presents the analysis of the vertex count distribution, separating the cases where the birth-death pair associated to the ring is located in region 1 or 2 on the persistence diagram. Figure 5c shows the distribution of pairwise distances between atoms in the ring structures. To analyze the distribution of pairwise distances within a-Si substructures with varying numbers of atoms, the histograms in Figure 5c are normalized to probability density. This approach differs from standard RDF or pair correlation functions.

The revised caption of Figure 5 is as follows:

*“Figure 5. Analysis of vertex counts and pairwise distances in ring structures. **a** Histograms showing the vertex counts in ring structures for the $N_{\text{max}10}$, $B_{\text{max}10}$, and $B_{\text{min}10}$ groups based on data from 10 samples. **b** Vertex count histograms for rings analyzed separately based on whether their birth–death pairs are in region 1 or 2. **c** Histograms of pairwise distances calculated using the atomic positions in the rings whose birth–death pairs are in region 1 or 2. To allow comparison between systems with different numbers of atoms, the histograms are normalized to the probability density.”*

Additionally, in line with the reviewer's suggestion based on the paper “Khalid Laaziri et al., Phys. Rev. Lett. 82 (1999) 3460”, we calculated the RDF and $S(k)$ for the a-Si system utilized in this study and included these results in the supplementary materials. As noted in our response to Comment 1-3 and Figure R2 above, our simulation data on RDF and $S(k)$ align reasonably well with the corresponding experimental data.

[Comment 1-6]

5. I cannot understand what “pairwise distances within ring structures” means.

[Our reply 1-6]

We appreciate the reviewer's feedback. The term "pairwise distances within ring structures" refers to distances calculated exclusively between atoms that belong to the ring structures identified through persistent homology. To clarify this point and prevent misunderstandings, we have included detailed explanations of the ring structure analysis in the caption of Figure 5, as discussed in our response to Comment 1-5 above.

Point-by-point response to the Reviewer #2:

[Comment 2-1]

The present manuscript of Minamitani et al. reports on the coupling between atomic-scale structures identified using so-called persistent homology and the intrinsic local elastic properties of amorphous Si. The authors generate a structure of amorphous silicon using molecular dynamics simulations to followingly analysis it by structural, mechanical, and persistent homology analysis. This brings them to identify a relation between atoms of large non-affine displacement which are found to have local structures of more vertices compared to atoms showcasing small non-affine displacements.

Generally, I think the work is of great interest to the community and has a significant potential in how to structurally identify regions of mechanical instability. However, I think the presented works lacks generality amongst more disordered/amorphous material families, fails to address important parts of the literature, and could greatly benefit from a more detailed analysis of the already presented data. Please see my detailed response below.

[Our reply 2-1]

We sincerely thank the reviewer for their thorough evaluation and for acknowledging the significance and potential impact of our work. We greatly appreciate the constructive feedback provided, which has been invaluable in improving the quality of our manuscript. We have carefully considered all the points raised and implemented the necessary revisions to enhance and strengthen the manuscript.

[Comment 2-2]

• Page 2: Stating that “The correlation between structure and properties of amorphous materials has long been a mystery in materials science, even for the fundamental mechanical properties” is to me a somewhat bold statement. Structure-property building in amorphous materials science has been ongoing for decades. This is evident in a number of cases with structure being correlated to a number of both thermal and mechanical properties, e.g. using coordination numbers or topological constraints as structural metrics for model-building. A few useful examples might be <https://doi.org/10.1016/j.commatsci.2018.12.004> and <https://doi.org/10.1016/j.jnoncrysol.2013.07.028>

[Our reply 2-2]

Thank you for your valuable comment and the references provided. We agree with the reviewer and acknowledge that our statement was somewhat exaggerated. In response, we have revised the statement and significantly rewritten the introduction.

To clarify the focus of this paper, we explicitly stated that it is limited to covalently bonded amorphous materials. While indicators such as coordination number, free volume, and bond orientation have successfully explained the correlation between local structure and softness in metallic glasses, it remains challenging to develop similar indicators for covalent amorphous solids due to the critical role of the atomic network structure. We introduced the references you provided as examples of previous efforts in this field, along with other representative studies.

Furthermore, we clarified that previous methodologies have struggled to address the objectives of this study. Specifically, we focus on elastic inhomogeneity, represented by the nonaffine term, and medium-range order (MRO). By utilizing our new analytical method based on persistent homology, we have revealed the correlation between these two factors and the local structure. This explanation has been added to the introduction to clearly highlight the novelty of this research.

[Comment 2-3]

• *Page 2, second paragraph: I think the authors overlook the recently published metric of softness (<https://doi.org/10.1126/science.aai8830>) – this metric is parameterized to quantify atoms as either soft or hard based on their local environment. While it is not a direct replacement of the suggested persistent homology method, the authors may consider discussing its relevance or even suing it as a source of comparison to e.g. their vertices counting.*

[Our reply 2-3]

Thank you for introducing the concept of softness (<https://doi.org/10.1126/science.aai8830>). To address the reviewer’s comment, we performed exploratory calculations of the following two-body $G(i; \mu)$ and three-body $\Psi(i; \xi, \lambda, \zeta)$ descriptors, which form the basis of the softness metric (Phys. Rev. Lett. 114, 108001 (2015)).

$$G(i; \mu) = \sum_j \exp(-(R_{ij} - \mu)^2 / L^2)$$

$$\Psi(i; \xi, \lambda, \zeta) = \sum_j \sum_k \exp\left(-\frac{R_{ij}^2 + R_{ik}^2 + R_{jk}^2}{\xi^2}\right) (1 + \lambda \cos \theta_{ijk})^\zeta$$

Using the hyperparameters reported in the above reference ($L=0.2$, $\mu=2.35$, $\lambda=1$, $\zeta=2$, and $\xi=4.7$), we compared these descriptors for three atomic groups in a-Si: NAmx10 (10 atoms with the largest nonaffine displacements), Bmin10 (10 atoms with the smallest Born terms), and Bmax10 (10 atoms with the largest Born terms). The results are shown in Figure R3.

As seen in Figures R3a and R3d, for the a-Si structure used in our study, the histogram distributions of $G(i; \mu)$ and $\Psi(i; \xi, \lambda, \zeta)$ are different for atoms in the NAmx10, Bmin10, and Bmax10 groups. However, the shape of histograms varied across different a-Si samples (Figures R3b and R3e). When averaged over 10 samples, the distribution differences among the groups become less distinct (Figures R3c and R3f). Applying machine learning techniques, such as SVM with optimized hyperparameters, could improve the general applicability of the softness metric for NAmx10, Bmin10, and Bmax10 in a-Si. Although this is beyond the scope of the current study, it will be considered in future work. We have cited the softness metric in the Discussion section as a valuable point of comparison for future research.

Figure R3: Calculation results of the two-body and three-body descriptors used in the softness metric

[Comment 2-4]

• Page 3 paragraph 1, please provide some examples of “conventional structural analysis methods”

[Our reply 2-4]

Thank you for your suggestion. Here, we define conventional structural analysis methods as those based on metrics such as coordination number, free volume, bond orientation, and related indicators. To highlight the distinctions between the persistent homology approach used in this study and conventional structural analysis methods, we have substantially revised the introduction. We have also provided more detailed explanations of the previous studies. The relevant section is in the following paragraph.

“Covalent amorphous materials present an even greater challenge for understanding structure–property correlations because these materials have complex network structures. Considering a method applicable to covalent systems, Demkowicz and Argon assessed whether the local environment of atoms is more liquid-like or solid-like based on interatomic angle distributions^{10,11}. Ma et al. defined the flexibility volume as the product of Voronoi volume and mean squared displacement^{12,13}. These two indicators were shown to correlate with the shear modulus in amorphous silicon (a-Si). Additionally, in silicate glass and hydrogenated amorphous silicon carbide, a correlation between average coordination number and hardness has been discussed based on topological constraint theory^{14,15}. These findings suggest that structural features at the atomic level affect mechanical properties in covalent amorphous materials, but two major challenges remain in establishing a clear connection between softness and nanostructure.”

[Comment 2-5]

• Page 3-5. I want to thank the authors for including a rather nice introduction to the concept of persistent homology. I think such paragraph is worthwhile for most readers as persistent homology is not yet well-established as a method in the materials science community. However, I think the paragraph also deserves to mention some of the drawbacks of the persistent homology method. I suggest the authors to at least discuss this.

[Our reply 2-5]

Thank you for your suggestion. In the revised manuscript, we have included a discussion on the drawbacks and challenges of persistent homology in the Introduction section.

One drawback of persistent homology is its inability to distinguish between structures with identical homological features. For instance, persistent homology cannot differentiate between fcc and hcp crystal structures when they share identical interatomic distances, as both produce identical hole sizes. However, in applications to disordered structures such as the amorphous materials analyzed in this study, this limitation has minimal impact.

Another challenge is that ring structures identified through inverse analysis as birth-death pairs are less intuitive compared to those derived from conventional ring statistics based on interatomic bonds. This is particularly problematic when discussing MRO. To address this, we utilized the concept of children in persistent homology to analyze the hierarchical structure of amorphous materials. In covalently bonded amorphous materials, MRO spans multiple scales, and the presence of children reflects this coexistence. This hierarchical relationship was key in interpreting MRO within the

framework of persistent homology.

The following section has been added to the introduction:

“From the above definitions, persistent homology offers the advantage of capturing structural features, such as rings and voids, at various scales simultaneously. Although persistent homology cannot describe fine differences in specific ordered structures, such as fcc and hcp crystals with the same interatomic distances, this limitation is less important in disordered systems. Consequently, persistent homology has been used to analyze structure and structure-property correlations in disordered materials³⁰⁻⁴¹.

Despite the successes of persistent homology in characterizing amorphous structures, directly linking persistent homology results to MRO remains challenging. A key difficulty is that the interpretation of the local structure assigned to each birth–death pair is less intuitive than that of conventional ring statistics based on atomic bonds. To address this, we also focus on the hierarchical information embedded in the persistence diagram. During a cycle transitioning to a boundary, smaller cycles may form and subsequently transition to boundaries within the larger cycle (red triangle in Figure 1d). In this paper, we refer to such smaller inner cycles as children, whereas this concept is referred to as a secondary ring in other reports³⁰.

The presence of children indicates that smaller-scale order exists within larger-scale order, corresponding to the coexistence of multiple scales of MRO observed in covalent amorphous materials. The blue and red polygons in Figure 1b provide examples of this correspondence. The red triangle, representing a child cycle, is associated with only two tetrahedra, reflecting the smallest scale MRO. In contrast, the blue hexagon, representing a parent cycle, is associated with three tetrahedra and corresponds to a larger-scale MRO. This demonstrates how the hierarchical structure in the persistence diagram captures the multiscale nature of MRO.”

[Comment 2-6]

• *Figure 1. Subfigure e is not referenced in the caption. Please provide a better and more elaborate caption text.*

[Our reply 2-6]

Thank you for your suggestion. We have revised the Figure 1 to include schematic image of key concepts in this study and updated the caption accordingly. The revised Figure 1 and its caption are shown below as Figure R4.

Figure R4 (revised Figure R1) Schematics of the key concepts and the method of persistent homology used in this study. **a** Visualization of nonaffine displacement in a three-dimensional amorphous structure under shear deformation, with a slice in the xy -plane. Gray circles denote atomic positions, and black arrows represent nonaffine displacements projected onto the xy -plane. **b** Illustration of a typical covalent amorphous network structure using a-Si as an example. SiSi_4 tetrahedral units share corners and medium-range order (MRO) defined by the angles between these tetrahedra. The blue and red polygons show examples of a parent cycle and its associated child observed in a covalent amorphous network. These cycles represent the hierarchical nature of MRO at different scales. **c–g** Filtration process in persistent homology with increasing sphere radii. Filled polygons represent cycles transforming into boundaries. The sequence of empty and filled polygons depicts the concept of children: the red triangle is a child of the blue pentagon, and the yellow triangle is a child of the green hexagon. **h** Persistence diagram obtained from the filtration process. The squared radii (\AA^2) were used as birth and death radii. Colors correspond to the polygons in

the filtration process.

[Comment 2-7]

• *Figures in general. Please provide larger text in the labels of the figures. Furthermore, why is there a difference in the amount of significant figures in the figure axes (even within the same subfigure)? For all figures please provide more descriptions of abbreviated variables in the captions – ideally to an extent where the figures can be understood independently from the main text.*

[Our reply 2-7]

We appreciate the reviewer’s feedback regarding the figures. In response, we have increased the font size of all graph labels and corrected inconsistencies in the significant figures on the birth and death axes of the persistence diagram. Furthermore, we have revised the captions to include detailed explanations of abbreviations and additional descriptions, ensuring that the figures can be understood independently of the main text.

[Comment 2-8]

• *Figure 2: I suggest the authors to color code the atoms in Figure 2b according to their Born and non-affine displacement. Perhaps as subfigures c and d?*

[Our reply 2-8]

Thank you for the suggestion. We have visualized the atomic structures with atoms color-coded based on the Born term (B_i) and non-affine displacement (D_i) as shown in the following Figure R5. These plots have been added as Figure S2 in the revised supplementary materials. In the main text (Figure 2a), the atomic structure is instead color-coded by the coordination number of each Si atom to emphasize the network connections.

Figure R5: a-Si structure models colored according to Born term B_i (left) and nonaffine

displacement D_i (right).

[Comment 2-9]

• Page 6 bottom: The authors choose to focus on only 10 atoms in each group. This is less than 0.1% of the atoms in the sample. While I understand it is indeed interesting to focus on the atoms of very high or low B_i and D_i , the authors should at least try to broaden their analysis to include more atoms, e.g. logarithmically by testing with 10, 20, 50, and 100 atoms. Simulations are somewhat prone to always have some extreme atoms which might not be very representative.

[Our reply 2-9]

We appreciate the reviewer's insightful comment, which enhances the validity of our discussion. To address this, we extended our analysis to include the top 20 and 50 atoms, as suggested. The results are presented in Figure S3 in the revised supplementary materials and shown below (Figure R6).

As the number of target atoms increases, the number of birth-death pairs rises correspondingly, making it more challenging to identify clear trends in the scatter plots. Nevertheless, analysis of the accumulated persistence diagrams, as suggested in Comment 2-13, revealed consistent trends in the birth-death pair distributions for ring structures involving atoms with high and low Born terms and large nonaffine displacements, similar to those observed for the top 10 atoms.

Furthermore, histograms of the number of vertices in the ring structures (Figure R7) showed similar trends regardless of whether we considered the top 10, 20, or 50 atoms. This consistency suggests that focusing on the top 10 atoms is sufficient to reasonably capture the overall behavior without losing critical information.

Figure R6: Dependency on the number of atoms considered in the analysis. **a–i** Birth–death pair distributions for the ring structures that include at least one atom from three groups: the top 10, 20, or 50 atoms with the largest nonaffine displacements (NAmax10, NAmax20, or NAmax50 in panels **a**, **e**, and **i**, respectively); the top 10, 20, or 50 atoms with the largest Born terms (Bmax10, Bmax20, or Bmax50 in panels **b**, **f**, and **j**, respectively); or the top 10, 20, or 50 atoms with the smallest Born terms (Bmin10, Bmin20, or Bmin50 in panels **c**, **g**, and **k**, respectively). **j–l** Accumulated persistence functions calculated from the birth–death distributions shown in panels **a–i**. The same figure is shown as Figure S3 in the revised supplementary materials.

Figure R7: Histograms of the number of vertices in ring structures analyzed separately for each group: rings containing at least one of the atoms characterized by large nonaffine displacement, large Born term, or small Born term. The terms NMax, Bmin, or Bmax are defined as in Figure S3. Analyses focusing on the top **a** 10, **b** 20, and **c** 50 atoms. The same figure is shown as Figure S5 in the revised supplementary materials.

[Comment 2-10]

• Page 7, top: Please elaborate further on what you mean by “inverse analysis”.

[Our reply 2-10]

Thank you for the suggestion. By “inverse analysis,” we refer to the process of identifying specific local atomic ring structures that correspond to particular birth–death pairs observed in the persistence diagram. To clarify this, we have added explanation as follows.

“In persistent homology, we can assign certain atomic ring structures to the respective birth–death pairs in a process called inverse analysis^{45–47}.”

[Comment 2-11]

• Page 7, top: In addition to the above comment, I assume this practically means going from an identified loop back to an atomic representation. As I guess the authors are aware this comes with a number of difficulties (or at least, active choices). In the methods you state that you employ the “stable volume” and “volume optimal cycle” in the inverse analysis. Please elaborate further on this in the main text in terms of what this specifically means for the identified loops and why you choose these over alternatives (shortest path, least atoms, etc. etc.).

[Our reply 2-11]

We thank the reviewer for this insightful comment. In response, we have elaborated the explanation in the Methods section.

As the reviewer correctly noted, “inverse analysis” refers to identifying candidate structures that correspond to a given birth-death pair. Selecting a single structure from homologically equivalent candidates can indeed be challenging. Several methods have been proposed for this purpose, including optimal cycles (Escolar, E. G., Hiraoka, Y.: Optimal Cycles for persistent homology via linear programming, Springer Japan, Tokyo, pp 79–96 (2016)), volume-optimal cycles (Obayashi, I. SIAM J. Appl. Algebra Geom. 2(4), 508–534 (2018).), persistent 1-cycles (Dey T. K., Hou T., Mandal S, Computational Topology in Image Context. Springer International Publishing, Cham, pp 123–136 (2019)), reconstruct shortest cycles (Čufar, M., J. Open Sour. Softw. 5(54), 2614 (2020).), and stable volumes (Obayashi, I. Journal of Applied and Computational Topology, 7, 671–706 (2023).).

Among these, we selected the stable volume and volume-optimal cycle methods for our analysis. Both methods can identify the children corresponding to birth-death pairs and aim to minimize the volume enclosed by the identified structure. The stable volume method is especially advantageous due to its robustness to noise, achieved by tightening the volume according to the noise strength, which provides more interpretable results compared to the others. However, due to algorithmic limitations, the stable volume method cannot be applied to pairs with small birth-death radius differences. For such cases, we used the volume-optimal cycle method as an alternative.

The following explanation was added to the Methods section:

“We identified the ring structure corresponding to each birth–death pair via the inverse analysis. In some cases, there are possible homologically equivariant candidate structures. To find the optimal structure from candidates, we applied the stable volume⁴⁷ and volume-optimal cycle⁴⁶ methods, both of which minimize the volume enclosed by the determined structure.

The stable volume method enhances robustness to noise by tightening the volume based on noise strength, providing more interpretable results. However, it cannot be applied to pairs close to the diagonal of the persistence diagram. For such cases, the volume-optimal cycle method was used instead.

For pairs for which death radius – birth radius > 0.1, the optimal structure was determined from the stable volume by setting the noise level to 0.001. For other cycles near the diagonal, the optimal structure was determined by using the volume-optimal cycle.”

[Comment 2-12]

• Page 7, top: You state that: "...that included Bmax10, Bmin10, and NAmx10". This needs elaboration. Does all 10 atoms in each group has to be in the loops for it to be analyzed? Just one of the atoms or maybe some?

[Our reply 2-12]

Thank you for your suggestion. In response, we have clarified the explanation to avoid misunderstandings.

We first identified ring structures that corresponds to all birth–death pairs within the interval [0, 5] for birth and death radii. From these, we selected ring structures that included at least one atom from the respective groups: Bmax10, Bmin10, or NAmx10. The revised explanation is as follows:

"We analyzed the ring structures corresponding to all birth–death pairs within the interval [0, 5] for birth and death radii. Then, we selected the ring structures that included at least one atom from the corresponding groups, Bmax10, Bmin10, or NAmx10. For example, if a ring structure contained an atom from NAmx10, it was classified into the NAmx10 group. The scatter plots of the birth–death pairs for the selected rings in each of these groups are shown in Figure 3c–e."

[Comment 2-13]

• Figure 3: I suggest making the points (or at least legends) larger and clearer. Furthermore, I wonder why the three datasets are all shown twice instead of just plotting them all in a single plot? I believe this division into three subfigures (3b-d) makes comparison harder.

[Our reply 2-13]

Thank you for this suggestion. To improve readability, we have increased the font size of the legends and data points in Figure 3. Regarding the layout of the figure, we considered combining the distributions of birth-death pairs for Bmin10, Bmax10, and NAmx10 into a single plot. However, we found that overlapping distributions in a combined plot would make it difficult to distinguish the characteristics of each group. At the same time, as the reviewer pointed out, the layout used in the initial version is redundant. Therefore, we have revised the figure to display each group in separate subfigures. This modification eliminates overlap and allows for clearer visualization and comparison of individual group characteristics.

The revised version of Figure 3 is shown below (Figure R8).

Figure R8 (revised version of Figure 3): Persistence diagram and birth–death pair distributions. **a** Persistence diagram derived from the structure in Figure 2a. **b** Accumulated persistence function (APF) for the birth–death pairs assigned to respective groups of atoms (Bmax10, Bmin10, and NAmx10). The classification of a birth–death pair into one of these groups is based on whether the corresponding ring structure obtained through inverse analysis contains at least one atom from the respective group. **c–e** Distribution of birth–death pairs for NAmx10 (**c**), Bax10 (**d**), and Bmin10 (**e**). The black square indicates the range of region 1, and the blue square indicates the range of region 2.

[Comment 2-14]

• *Figure 3 and related discussion: Often, persistence diagrams are very qualitative and difficult to interpret. I suggest the authors to use more quantitative measures for drawing their conclusions, e.g. the recently developed accumulated persistence function, see.: <https://doi.org/10.1080/10618600.2019.1573686>*

[Our reply 2-14]

Thank you for your suggestion. We applied the accumulated persistence function (APF) to quantitatively evaluate the distribution of birth–death pairs of the rings that include at least one atom in NAmx10, Bmin10, or Bmax10 groups.

Since the number of birth-death pairs of the three groups are different, here we define the normalized version of APF defined as follows:

$$APF_G(m) = \frac{1}{|G|} \sum_{i:m_i < m} (d_i - b_i)$$

Here, the subscript G identifies the group of NMax10, Bmin10, or Bmax10. $|G|$ is the number of birth-death pairs in each group. b_i and d_i are the birth and death radii at i -th birth-death pair and $m_i = (d_i - b_i)/2$

The results together with the above definition have been added in the main text and Figure 3b (Figure R8b). Additionally, to address Comment 2-9, we calculated the APF for cases where the number of atoms of interest was increased (e.g., 20, 50 atoms). As shown in Supplementary Figures S3–S5, the characteristics of the birth–death pair distributions remain consistent regardless of the number of atoms considered. This confirms the robustness of our conclusions.

[Comment 2-15]

• *Figure 5: I suggest the authors also to compare between the graphs vertically. While some differences between graphs are clear (e.g. in Fig. 5a) others are more subtle (e.g. 5c).*

[Our reply 2-15]

We appreciate the reviewer’s feedback and suggestion to explore vertical comparisons between the graphs.

In Figure 5a, we compare the number of vertices in the ring structures that include at least one of Bmin10, Bmax10, or NMax10, revealing clear differences between the three groups. However, the primary purpose of Figures 5b and 5c is to examine the differences within each group when the corresponding birth–death pairs are located in region 1 or region 2. As expected, the size distribution and shape of the ring structures in region 1 and region 2 exhibit similar trends across all groups.

To address the reviewer’s suggestion, we explored a vertical comparison of the graphs in Figure 5b. This attempt, shown as Figure R9 below, demonstrates consistent trends across the groups and provides further insight into the subtle differences observed in Figures 5b and 5c. While this additional comparison offers valuable context, we believe the current presentation effectively conveys the main findings.

Figure R9: Histograms of the number of vertices in the ring structures including at least one atom in three groups: NAmx10 (top 10 atoms with large non-affine displacement), Bmax10 (top 10 atoms with large Born term), or Bmin10 (top 10 atoms with small Born term). The left panel shows the results for rings whose corresponding birth–death pairs are located in region1, while the right panel shows those in region2.

[Comment 2-16]

• Page 9, second paragraph: *The authors identify region 1 and region 2 to have very high and low vertex counts, respectively. Given previous literature on oxides (Refs. 20 and 26 in the paper), this is hardly any surprise as longer rings with more atoms will inherently be found in low-birth, high-death regions (such as region 1). Please discuss further and provide more elaborate referencing to existing literature.*

[Our reply 2-16]

Thank you for your valuable suggestion. As the reviewer correctly pointed out, the observation that ring structures with larger death radii are associated with high vertex counts has been previously reported in the literature. To provide proper context, we have added references to prior studies discussing this trend (Refs. 30,32,33 in the revised manuscript).

The primary focus of this section lies in understanding how the birth-death pairs in region 1 correspond to MRO, rather than merely the ring size differences. To address this, we analyzed not only the vertex count distribution but also the pairwise interatomic distance distributions for rings in regions 1 and 2. We have revised the manuscript accordingly, as follows:

“We also analyzed the vertex counts separately of the ring structures whose birth–death pair positions

were in regions 1 and 2 (Figure 5b). As expected from the differences in death radii values, the vertex counts for the rings whose birth–death pairs are in region 2 were small, whereas the vertex counts for those in region 1 exhibited a long-tail distribution. A similar trend has been discussed in previous studies^{30,32,33}.

In addition to this simple difference in ring size, the rings in these two regions also made distinct contributions to MRO, as shown in the histogram of pairwise interatomic distances. The histogram of pairwise distances between atoms forming ring structures whose birth–death pair positions were in region 1 revealed a sharp first peak and a clear second peak near 3.8 Å, which corresponded to small-scale MRO (Figure 5c). The position of this second peak aligned with the death radius distribution in region 1, estimated as follows: when the death radius was 4.0 Å², the radius of spheres in the filtration was 2.0 Å, leading to an interatomic pairwise distance of 4.0 Å. In contrast, the histogram for region 2 showed a broader first peak and an additional peak near 3.2 Å, which was absent in region 1. The presence of this additional peak indicated short-range disorder in bond lengths and angles, disrupting the MRO.”

[Comment 2-17]

• Page 12, bottom: I suggest the authors to introduce and discuss the use of Eq. 12 in the main text and not in the Methods section

[Our reply 2-17]

We appreciate the reviewer’s suggestion. In response, we have introduced the equation for the decomposition of nonaffine displacement at the beginning of the “Correlation with low-energy localized vibrational excitations” section in the main text. The revised text is as follows:

“The nonaffine displacement can be decomposed by the vibrational mode eigenvector, ψ_p .^{16,50}

$$\mathbf{D}_{i,\alpha\beta}^{NA} = \frac{\partial \mathbf{r}_i}{\partial \eta_{\alpha\beta}} = \frac{1}{\lambda_p} \sum_{j \neq i} \sum_p \left(\frac{\partial^2 U}{\partial \mathbf{r}_j \partial \eta_{\alpha\beta}} \cdot \psi_p \right) \psi_p. \quad (2)$$

Here, U is the potential energy of the system, $\eta_{\alpha\beta}$ is the strain, α and β are the indices specifying direction x, y , or z , and i and j are indices specifying the atoms. λ_p is the p -th eigenvalue of the Hessian matrix.”

[Comment 2-18]

• Page 13, top: The authors state “... for 1000 low-energy vibrational modes”. Do you mean the 1000

modes of lowest eigenfrequencies? What frequency range does this correspond to?

[Our reply 2-18]

We appreciate the reviewer's feedback. As the reviewer correctly pointed out, we focused on the IPR of the 1000 vibrational modes with the lowest eigenfrequencies. To clarify this, we have added a plot of the IPR as a function of frequency, illustrating the frequency range of these modes.

As shown in Figure R10, the frequency range spans from 0 to approximately 2.5 THz. This additional information is included in the revised supplementary materials as Figure S7.

Figure R10: Calculation results for inverse participation ratio (IPR). IPR for all vibrational modes as a function of **a** mode index and **b** frequency. **c** and **d** are magnified views of panels **a** and **b**, focusing on the 1000 lowest frequency modes. The red dashed line indicates the threshold for identifying low-energy localized modes ($IPR = 50$), whereas the gray dashed lines represent the upper limit ($IPR = N_{atom} = 13,824$) and lower limit ($IPR = 1$) of the IPR.

[Comment 2-19]

- *Page 13: A minor thing, but IPR is usually coined as “inverse participation ratio” (not inversion).*

[Our reply 2-19]

Thank you very much for your careful review. We have corrected the corresponding part in the text to accurately reflect the term as follows:

“Figure 8a shows the calculation of the inverse participation ratio (IPR) for 1000 modes with lowest frequencies.”

[Comment 2-20]

- *Page 13: I suggest the authors to provide the fundamental limits of the IPR (1 and N) as well as provide a plot of the participation ratio (that is, the inverse of IPR) – at least in the SI. Usually, highly localized modes are given as modes with a $PR < 0.15$ but it is currently very difficult to judge exactly how localized the studied modes in fact are.*

[Our reply 2-20]

Thank you for your suggestion. To address this, we have explicitly provided the lower and upper limits of the IPR in Figure R10 and Figure S7. Additionally, we calculated the participation ratio (PR) and plotted it as a function of both mode index and mode frequency (Figure R11).

To identify highly localized modes, we highlighted modes with $IPR > 50$, which we used as the criterion for localization. As shown in Figure R11, all these modes have $PR < 0.1$, which aligns with the criteria suggested by the referee. The same figure has been included as Figure S8 in the revised supplementary materials.

Figure R11: Calculation results of participation ratio (PR). PR for all vibrational modes as a function of **a** mode index and **b** frequency. Panels **c** and **d** are magnified views of **a** and **b**, focusing on the 1000 lowest frequency modes. The red circles indicate modes for which IPR > 50.

[Comment 2-21]

• *Eq. 2 please provide a description of the lambda variable.*

[Our reply 2-21]

Thank you for the suggestion. We have included the description of the lambda variable at the beginning of the “Correlation with low-energy localized vibrational excitations” as written in the response to Comment 2-17.

[Comment 2-22]

• *Page 14: I suggest to significantly deepen the analysis discussion on the connection between the*

identified structural features and vibrational properties. Currently it seems somewhat vague and superficial. One could test e.g. the correlation between B_i , D_i , and IPR and possibly show in the SI. For more, the very low-frequency modes are responsible for a large amount of literature which seems to be somewhat overlooked, e.g. in relation to the so-called boson peak, see e.g. <https://doi.org/10.1103/PhysRevLett.117.035501>.

[Our reply 2-22]

We appreciate the reviewer's feedback, which has greatly helped us refine our analysis and discussion. Following the reviewer's suggestion, we investigated the correlations among Born term B_i , nonaffine displacement D_i , and IPR. To achieve this, we introduce the particle-based inverse participation ratio (IPR_i) as

$$IPR_i = N_{atom} \sum_{p=1}^{1000} (\psi_{p_{ix}})^4 + (\psi_{p_{iy}})^4 + (\psi_{p_{iz}})^4,$$

where $\psi_{p_{ix}}$, $\psi_{p_{iy}}$, and $\psi_{p_{iz}}$ denote the x , y , and z components of the p -th eigenvector, $\boldsymbol{\psi}_p$. IPR_i quantifies the degree of vibrational localization for each particle i . In Figure R12 below, we plot D_i against IPR_i and B_i against IPR_i . Figure R12a clearly shows a correlation between D_i and IPR_i , suggesting that particles with large nonaffine displacements are associated with highly localized vibrational modes. In contrast Figure R12b indicates no correlation between B_i and IPR_i . These findings lead us to conclude that localized modes are associated with large nonaffine displacements but not with small Born terms.

Figure R12: Correlation between particle-based inversion participation ratio (IPR_i) and **a** nonaffine displacement D_i and **b** Born term B_i .

We have also addressed the reviewer's suggestion to discuss localized modes in relation to the boson peak. Previous works have consistently demonstrated a strong connection between localized modes and the boson peak. Notably, Shimada et al. (Phys. Rev. E 98, 060901(R) (2018)) showed that the localized modes arise from the boson peak, with their frequencies reduced by repulsive interactions. Additionally, Moriel et al. (Phy. Rev. Research 6, 023053 (2024)) revealed that the boson peak comprises multiple spatially coupled localized modes. Hu and Tanaka (Nature Physics 18, 669 (2022)) reached a similar conclusion. These studies indicate that the localized modes and the boson peak share a common origin. By integrating these insights with the correlation between localized modes and MRO revealed in our work, we suggest that the structural origin of the boson peak can be attributed to MRO.

In the revised manuscript, we have added the data in Figure R12 as Figure S9 in the supplementary materials. Additionally, we have included the above discussion on localized modes in connection with the boson peak and their relation to MRO in the Discussion section.

[Comment 2-23]

• Page 18: You write: “, which is consistent with previous reports.” Please provide suitable references for this statement.

[Our reply 2-23]

Thank you for your suggestion. We have added reference 43 (Talati, M., Albaret, T. & Tanguy, A. Atomistic simulations of elastic and plastic properties in amorphous silicon. *EPL* **86**, 66005 (2009)) in the revised manuscript.

[Comment 2-24]

• Generally, I think the biggest caveat of the study comes to how only amorphous Si is tested. While I acknowledge that amorphous Si is a very thoroughly tested and good reference material it is also one of the absolute simplest disordered materials both structurally and dynamically. As such, while the present study serves somewhat as a proof of concept, the lack of generalization to other amorphous systems (oxides, chalcogenides, metals, halides, nitrides, all possibly with multiple components) significantly dampens the possible applicability of the suggested method.

[Our reply 2-24]

Thank you very much for your valuable suggestion. We fully agree that it is crucial to consider the applicability of our method to other materials, as the reviewer has pointed out. In response, we have added a discussion on how the proof of concept in this study using a-Si can be generalized to other amorphous systems. The added discussion is as follows:

“The versatility of our persistent homology-based analysis extends beyond a-Si. For example, AX₂ covalent amorphous solids, such as a-SiO₂ and a-GeSe₂, are promising targets for similar studies because of their well-defined tetrahedral network structures²⁶. By focusing on the tetrahedral units, the approach used in this study could be readily adapted to these systems. Moreover, binary metallic glasses, such as Pd-Ni and Cu-Zr alloys, are intriguing comparative targets. These materials contain polyhedral clusters as structural units, suggesting that our method could be applicable. However, it is necessary to address challenges including determining whether first-order or second-order diagrams are more suitable for capturing structural features^{30,62,63} and how to extend persistent homology to consider multiple elemental species. Combining these analyses with high-accuracy machine-learning potentials⁶⁴⁻⁶⁶ holds great promise for exploring key correlations between hierarchical structures, MRO, nonaffine deformation, and localized vibrations across various glassy materials. ”

Point-by-point response to the Reviewer #3:

[Comment 3-1]

This paper studies local structures induced by Born terms and large non-affine terms by means of persistent homology analysis, and furthermore, tries to find a relationship with low-energy localized vibrational excitations. The analysis tools they present look powerful to identify subtle but significant atomic structures to distinguish those two terms. The computational code of the persistent homology analysis (HomCloud) is open to the public, so the reader can easily follow their analysis to other materials. I believe that the proposed analysis and results will be of interest to a broad audience of the journal. Therefore, I basically would like to accept this paper for publication, under the condition that the authors take care of the following concerns and suggestions as detailed below.

[Our reply 3-1]

We thank the reviewer for their time spent reviewing our manuscript. We are also sincerely grateful for the reviewer's positive evaluation of our research and their encouraging comments. As requested, we have carefully addressed all the concerns and suggestions provided by the reviewer to enhance the quality of our manuscript. We believe these revisions have strengthened the clarity and impact of our findings.

[Comment 3-2]

I think that the result between line-191 and line-201 is significantly important for the discussions afterwards, but the discussion here seems to be not quantitative. I strongly recommend the authors to modify this part so that the conclusions will be derived in more quantitative way. One idea may be to use some distance on persistence diagrams to measure the similarity/difference.

[Our reply 3-2]

Thank you for the insightful suggestion. As the reviewer correctly pointed out, the original manuscript lacked a quantitative analysis of the differences in the distributions of birth-death pairs among B_{min10} , B_{max10} , and N_{Amax10} . To address this, we employed the accumulated persistence function (APF), as suggested by Reviewer 2, to quantify the differences among these three groups. The calculation results were added as Fig. 3b (the revised Figure 3 is shown as Figure R8 in this file). The updated and more quantitative discussion in the revised manuscript is presented below:

“The distribution of birth–death pairs differed substantially between the B_{min10} and N_{Amax10} groups, clearly indicating that the structural features captured by persistent homology also differed. To evaluate the difference in the distribution of birth–death pairs quantitatively, we used the accumulated

persistence function (APF)³², defined as follows.

$$APF_G(m) = \frac{1}{|G|} \sum_{i:m_i < m} (d_i - b_i) \quad (1)$$

Here, subscript G identifies the N_{max10} , B_{min10} , or B_{max10} group. $|G|$ is the number of birth-death pairs in each group. b_i and d_i are the birth and death radii for the i -th birth-death pair and $m_i = (d_i - b_i)/2$.

The APF results highlighted the differences in the birth-death pair distributions among the B_{min10} , B_{max10} , and N_{max10} groups. All three APFs exhibited a step-like shape, with the initial rise near $m = 2.3$ and saturation around $m = 3.0$. Although the regions where APF changed were the same among the groups, the saturation values differed markedly. Despite slight sample-to-sample variation, the saturated APF values consistently followed the order $B_{max10} > N_{max10} > B_{min10}$. This result implied that the minimum difference between death and birth radii was consistent among the groups. However, the number of pairs with a large difference, namely, pairs far from the diagonal, varied in this order.

This observation was consistent with the tendencies observed in the scatter plots of the birth-death pairs. For the B_{max10} and N_{max10} groups, birth-death pairs were primarily found in region 1 ([1.3, 1.7] and [3.3, 4.8] for birth and death radii, respectively). In contrast, region 2 ([1.7, 2.2] and [2.7, 3.2]) contained fewer pairs. The B_{min10} group exhibited the opposite trend. Similar trends persisted regardless of whether we analyzed the top 20 or top 50 atoms or examined 10 samples (Figure S3 and S4). Therefore, we concluded that the local structures of B_{max10} and N_{max10} were characterized by birth-death pairs in region 1, whereas B_{min10} was characterized by those in region 2.”

[Comment 3-3]

The discussion from line-236 to line-250 should be revised more logically. The point I am not satisfied is that, since the definition of SRO and MRO in this paper is ambiguous (at least for me), the assignment of SRO and MRO to the local structures looks ambiguous and less convincing. I would like to understand this part more by adding further logical and quantitative explanations.

[Our reply 3-3]

Thank you for your insightful comment. The ambiguity in the definitions of SRO and MRO in a-Si has been also pointed out by Reviewers 1, 2, and 4. To address this, we have substantially revised the introduction to clarify these definitions.

a-Si has a network structure composed of tetrahedral units, where each silicon atom is coordinated by four surrounding silicon atoms. The constraints in the distribution of Si-Si bond lengths correspond to short-range order (SRO). The way these tetrahedra are connected determines the medium-range order (MRO). The smallest scale of MRO is defined by whether adjacent tetrahedra share a corner, an edge, or a face. As more tetrahedra—three, four, or more—connect together, the definition of MRO extends to larger scales. We have summarized these general definitions of SRO and MRO in covalent amorphous materials and explained the existence of multiple MRO scales in the revised introduction as follows:

“Generally, MRO refers to the atomic organization on a scale of 5 to 20 Å that does not appear in purely random configurations^{26,27}. In covalent amorphous materials, network structures are formed by polyhedral units created by atomic bonds, which share corners, edges, or faces. Figure 1b shows an example of a corner-sharing network. The organization of the bond-length scale within polyhedral units is called short-range order (SRO), whereas MRO corresponds to the relative angles and connectivity between polyhedra^{26,27}. The angle formed between two adjacent polyhedra corresponds to the dihedral angle formed by four atoms, the restriction of which represents the smallest MRO scale. When angles among multiple polyhedra are correlated, they constrain the atomic ring structures and impose MRO on a longer scale. However, the relationship between MRO in these covalent amorphous networks and softness remains unexplored.”

Based on these definitions of SRO and MRO, we have revised the explanation for the section corresponding to lines 236 to 250 as follows:

“We also analyzed the vertex counts separately of the ring structures whose birth–death pair positions were in regions 1 and 2 (Figure 5b). As expected from the differences in death radii values, the vertex counts for the rings whose birth–death pairs are in region 2 were small, whereas the vertex counts for those in region 1 exhibited a long-tail distribution. A similar trend has been discussed in previous studies^{30,32,33}.

In addition to this simple difference in ring size, the rings in these two regions also made distinct contributions to MRO, as shown in the histogram of pairwise interatomic distances. The histogram of pairwise distances between atoms forming ring structures whose birth–death pair positions were in region 1 revealed a sharp first peak and a clear second peak near 3.8 Å, which corresponded to small-scale MRO (Figure 5c). The position of this second peak aligned with the death radius distribution in region 1, estimated as follows: when the death radius was 4.0 Å², the radius of spheres in the filtration was 2.0 Å, leading to an interatomic pairwise distance of 4.0 Å. In contrast, the histogram for region

2 showed a broader first peak and an additional peak near 3.2 Å, which was absent in region 1. The presence of this additional peak indicated short-range disorder in bond lengths and angles, disrupting the MRO.

Based on these results, we concluded that the birth–death pairs in region 2 corresponded to structures with short-range disorder; whereas those in region 1 corresponded to MRO structures. Similar distributions of birth–death pairs and their correlations with short-range disorder and MRO formation have been previously reported³⁶.”

[Comment 3-4]

Unfortunately, the quality of English seems to be not satisfactory. I will not raise all those parts here, but I recommend the authors to seriously recheck English in the manuscript.

[Our reply 3-4]

We thank the reviewer for pointing out the importance of language quality. We take this comment very seriously and have revised and thoroughly edited the entire manuscript for language. Additionally, we had the manuscript reviewed by a native English speaker.

Point-by-point response to the Reviewer #4:

[Comment 4-1]

The manuscript presents molecular dynamics simulations of a 13,824-atom structure of amorphous silicon, described with the Stillinger-Weber potential [Phys. Rev. B Condens. Matter 31, 5262–5271 (1985)]. Molecular dynamics is used to compute the affine and non-affine components of the atomic displacements from equilibrium. Then, the authors look for a correlation between persistent homology analyses for atoms having particularly large or small affine and non-affine displacements.

The manuscript further attempts to connect hierarchical structures identified by the persistent homology analysis with low-energy localized vibrational excitations. To the best of my understanding, this connection is the novel part of the work, but I am not convinced this meets the standards for publication in Nature Communications for the following reasons:

[Our reply 4-1]

We sincerely thank the reviewer for their time and effort in reviewing our manuscript. We greatly appreciate the detailed feedback, which has been invaluable in refining the manuscript and addressing key points of improvement. Below, we respond to the concerns and suggestions raised by the reviewer and provide detailed explanations of the revisions made. We hope that the revised manuscript meets the reviewer's expectations.

[Comment 4-2]

1) The main messages of the manuscript are not clearly explained. The manuscript is written in a style suitable for a highly specialized journal, not a high-impact journal such as Nature Communications. To the best of my understanding, there are two main messages in this work:

A) The existence of a relation between low-energy localized vibrational excitations and hierarchical structures emerging from the persistent homology analysis;

B) The connection between atomic displacements, structures emerging from the persistent homology analysis, and elastic moduli.

These points are not clearly explained, and only an expert reader familiar with the field can extrapolate this after reading the manuscript multiple times. For example, all the details on the elastic moduli are reported in the methods, and there is no figure to highlight the connection between structures emerging from the persistent homology analysis and elastic moduli.

[Our reply 4-2]

We are grateful for the reviewer's insightful feedback, which has helped us improve our manuscript to appeal to a broader audience. In response to this comment, we have significantly revised the introduction and discussion sections to clearly highlight the key points (A) and (B) earlier and in a more accessible manner compared to the previous version. Specifically, we have implemented the following revisions, which we believe make the manuscript more approachable to non-specialists:

1. We added a detailed explanation of the concept of elasticity of glasses, carefully tailored for general readers. In particular, we have clarified the concept of nonaffine displacements.
2. We provided a clear explanation of the vibrational modes of glasses, with a focus on localized vibrational modes and the boson peak phenomenon.
3. In response to a suggestion from Reviewer 1 (Comment 1-2), we included an introductory discussion of amorphous silicon and its medium-range order (MRO).
4. We clearly explained our main findings (A) and (B) in the introduction section. In particular, we have highlighted that both nonaffine displacements and localized vibrations correlate with MRO, emphasizing the contributions of this study.
5. To complement the textual explanation, we included schematics (Figure 1a and 1b in the revised manuscript, which are also shown as Figure R4a and R4b in this file) to illustrate key concepts such as nonaffine displacement and MRO.

We hope these revisions address the reviewer's concerns and make the manuscript more accessible and impactful.

[Comment 4-3]

2) One of the most important claims made by the authors (1A) involves the localization of vibrational modes in disordered systems. As discussed in [H. R. Schober and C. Oligschleger PRB 53 1996 <https://link.aps.org/doi/10.1103/PhysRevB.53.11469>], and in more recent articles that cite this work, the localization properties of low-frequency vibrational modes in disordered systems are subject to spurious, non-physical effects caused by periodic boundary conditions. The article above discusses a procedure to correct for these spurious effects.

The authors simulate a structure containing 13,824 atoms using molecular dynamics in periodic

boundary conditions. They do not provide evidence that their results (e.g. Fig. 8) are physically meaningful and not affected by these spurious finite-size effects.

[Our reply 4-3]

We appreciate the reviewer for their important comments. As the reviewer commented, we expect finite system-size effects in our results of the localized modes, which arise from hybridizations with spatially extended phonon modes. To eliminate these effects, we need to address the hybridizations using the method described in “Schober and Oligschleger, Phys. Rev. B 53, 11469 (1996)” (paper suggested by the reviewer) or the more recent technique presented in “Shcheblanov et al., Phys. Status Solidi B 258, 2000422 (2021).” However, this hybridization primarily affects the vibration in the region far away from the core region where the amplitude of localized mode is large. Our analysis in Figure 8 focuses on the localized core region in relation to the nonaffine displacements, which should remain unchanged. Therefore, we conclude that our findings and conclusions are not influenced by finite system-size effects (hybridizations).

In the revised manuscript, we have added one paragraph which includes the preceding discussions on hybridization and the finite-size effects in the localized modes.

[Comment 4-4]

3) The computational methodology employed is outdated, based on the Stillinger-Weber potential [Phys. Rev. B Condens. Matter 31, 5262–5271 (1985)] developed 39 years ago. The authors could have done the same analysis using state-of-the-art and quantum-accurate machine-learning potential. The persistent homology analysis based on the HomCloud software is useful but not novel. The novelty of the present work stands in the relation between structural motifs emerging from the persistent homology analysis and elastic moduli, but as mentioned at point 1 this is not clearly explained.

[Our reply 4-4]

We appreciate the reviewer for their insightful comments. Based on their suggestions, simulating a-Si using machine-learning (ML) potential could be a significant future endeavor. In our current work, we utilized the Stillinger-Weber (SW) potential, which allows us to derive analytical expressions for the Born terms, nonaffine displacements, and Hessian matrix, since the SW potential provides analytical forms for derivatives of the potential. However, the ML potential presents a more complex challenge to calculate its derivatives, meriting consideration for future research, but it lies outside the scope of our current study.

We concur with the reviewer that the persistent homology analysis using the HomCloud software is

not novel. The innovation of this study lies in our application of persistent homology, specifically targeting soft spots characterized by small Born terms and significant nonaffine displacements. This approach allows us to identify structural features in the soft spots. As mentioned in our reply 4-2, we have clarified this point in the revised manuscript, incorporating an additional figure (Figure 1a, b) to illustrate it more clearly.

Additionally, we wish to highlight that HomCloud software is publicly accessible, allowing readers to easily follow the current analysis. This could draw the attention of a wide audience in Nature Communications, as also noted by reviewer 3.

Following the referees' comments, in the revised manuscript, we have included the comments on the ML potential in the discussion section as follows.

“Combining these analyses with high-accuracy machine-learning potentials^{64–66} holds great promise for exploring key correlations between hierarchical structures, MRO, nonaffine deformation, and localized vibrations across various glassy materials.”

[Comment 4-5]

Minor comments:

4) Could the author clarify and expand the following sentence?

“Because homology groups are defined as the quotient group of cycles by boundaries, the transformation from cycles to boundaries is fundamentally important.”

In particular, what do they mean by “the transformation from cycles to boundaries is fundamentally important”?

[Our reply 4-5]

Thank you very much for this feedback. The transformation from cycles to boundaries corresponds to the disappearance of holes in the topological space. In the revised manuscript, we have rewritten the explanation of persistent homology using simpler and more accessible language. The revised section reads as follows:

“Motivated by these challenges, the present work aims to clarify the correlation between nonaffine displacements, low-energy localized vibrations, and MRO based on concepts from mathematical topology, that is, persistent homology^{28,29}. Schematics of the calculation procedure are shown in Figures 1c–g. Imagine a sphere is placed at each atom, with the radius gradually increasing. When two spheres touch, then a bond is assigned between the atoms. As the radius increases, the number of

bonds grows, and at a certain point, polygonal rings are formed. These rings correspond to one-dimensional holes in the topological space and are referred to as cycles. As the sphere radius continues to increase, the polygons become covered by the spheres. At this point, the cycle has transformed into the boundary of a filled polygon, meaning that the hole no longer exists.”

[Comment 4-6]

5) Which message are the authors trying to convey with Fig. 2b? If the message is that the structure is disordered, this is not clearly conveyed in Fig. 2b.

[Our reply 4-6]

Thank you very much for this feedback. The message we wanted to convey with Figure 2b in the original manuscript is the characteristics of atomic structure in a-Si. To highlight this point, we colored Si atoms by coordination number and show the bond between atoms in the revised Figure 2a. The same figure is shown in below (Figure R13).

Figure R13. Structure model of amorphous Si and calculated Born term (B_i) and nonaffine displacement (D_i) for each atom in the model. **a** Amorphous structure comprising 13,824 Si atoms used for the analysis. The white, red, and blue atoms represent 4-, 5-, and 3-coordinated Si atoms, respectively. **b** Scatter plot of D_i and B_i .

[Comment 4-7]

6) The authors mention multiple times "scale of short-range order (SRO)" or "scale of medium-range order (MRO)", but do not clearly quantify such scale. Could the authors elaborate that more clearly, and highlight (ideally in a figure, e.g., in an extension of Fig. 1) the connection between the persistence diagram and SRO/MRO scale.

Thank you for your insightful comment. The ambiguity in the definitions of SRO and MRO in a-Si

has been pointed out by Reviewers 1-3. To address this, we have substantially revised the introduction to clarify these definitions.

a-Si has a network structure composed of tetrahedral units, where each silicon atom is coordinated by four surrounding silicon atoms. The constraints in the distribution of Si-Si bond lengths correspond to short-range order (SRO). The way these tetrahedra are connected determines the medium-range order (MRO). The smallest scale of MRO is defined by whether adjacent tetrahedra share a corner, an edge, or a face. As more tetrahedra—three, four, or more—connect together, the definition of MRO extends to larger scales. We have summarized these general definitions of SRO and MRO in covalent amorphous materials and explained the existence of multiple MRO scales in the revised introduction as follows:

“Generally, MRO refers to the atomic organization on a scale of 5 to 20 Å that does not appear in purely random configurations^{26,27}. In covalent amorphous materials, network structures are formed by polyhedral units created by atomic bonds, which share corners, edges, or faces. Figure 1b shows an example of a corner-sharing network. The organization of the bond-length scale within polyhedral units is called short-range order (SRO), whereas MRO corresponds to the relative angles and connectivity between polyhedra^{26,27}. The angle formed between two adjacent polyhedra corresponds to the dihedral angle formed by four atoms, the restriction of which represents the smallest MRO scale. When angles among multiple polyhedra are correlated, they constrain the atomic ring structures and impose MRO on a longer scale. However, the relationship between MRO in these covalent amorphous networks and softness remains unexplored.”

We also added the visualization of the above point as Figure 1b in the revised manuscript. The same figure is shown as Figure R1 in this file.

We would like to thank all the reviewers for their constructive and insightful comments, which have significantly improved the quality of our manuscript. For clarity, all modifications in the main manuscript and supplementary materials have been highlighted using grey markers.

Below, we provide point-by-point responses to each of the reviewers' comments.

Responses to Reviewer 1

General remark

The author has made significant revisions to the original text. This is highly commendable. However, there are serious problems with the structural model created by the MD simulations that warrant reconsideration. I am also afraid that the structural information obtained from the PD analysis is too specialized and that this paper would be better suited for publication in a more specialized journal. Therefore, we cannot accept this paper in its present form.

Our response

We appreciate the reviewer's time and thoughtful comments on our manuscript. However, we would like to clarify that the primary objective of our research is not to precisely replicate the structural features of experimental amorphous silicon (a-Si) but rather to elucidate the local structural origins of inhomogeneous nonaffine displacements in covalent networks using persistent homology. While structural comparisons provide valuable context, we argue that minor deviations in the structure factor and radial distribution function do not compromise the validity of our findings, as detailed below.

As a representative model of amorphous structures with covalent networks, we utilize a-Si structures generated via melt-quench simulations based on the Stillinger-Weber potential. The following references have employed similar simulations to generate a-Si structures for investigating the properties of covalently bonded amorphous materials, and these approaches are widely accepted within the physics community:

- M. J. Demkowicz, A. S. Argon, High-density liquidlike component facilitates plastic flow in a model amorphous silicon system. *Phys. Rev. Lett.* 93, 025505 (2004).
- C. Fusco, T. Albaret, A. Tanguy, Role of local order in the small-scale plasticity of model amorphous materials. *Phys. Rev. E* 82, 066116 (2010).

Furthermore, in the revised manuscript, we confirm that the same conclusions hold even when varying the parameters within the Stillinger-Weber potential, which modifies the structure and network characteristics (see *Supplementary materials, “Dependence of structural and mechanical properties on tetrahedrality parameter in Stillinger–Weber potential”*). Based on this, we are confident that the minor structural deviations from experimental results pointed out by the reviewer do not compromise the validity or reliability of our analysis and conclusions.

The most significant contribution of this paper is the application of persistent homology to reveal, for the first time, the hierarchical nature of local structures governing nonaffine displacement and their correlation with low-frequency localized vibrational modes. This type of analysis is increasingly recognized as a valuable approach in the study of amorphous and glassy materials, and we respectfully disagree with the assessment that it is “too specialized.”

Below, we address each specific comment in detail.

Comment 1

The agreement between the experimental $S(Q)$ and $g(r)$ and computed ones in comparison with data reported by V. L. Derin et al, *Nature*, 589, 59 (2021)(GAP-MD)and in “Computer Simulations of Glasses: Methodologies and Applications”, Ed. by J. Du and A. N. Cormack, Wiley-American Ceramic Society, Hoboken (2022) pp. 60 (MD-RMC). I am really afraid that we cannot see good agreement even when the Q_{\max} value is extended to 20 \AA^{-1} .

Our response 1

The references cited by the reviewer differ significantly in both their objectives and simulation methodologies from our study.

• **Deringer et al., *Nature*, 589, 59 (2021):** This study focused on the development of machine-learning potentials capable of tracking phase transitions in a-Si under high-pressure conditions.

• **J. Du and A. N. Cormack (2022):** We could not locate a book titled *Computer Simulations of Glasses: Methodologies and Applications*. It is possible that the reviewer was referring to *Atomic Simulation of Glasses: Fundamentals and Applications*. edited

by J. Du and A. N. Cormack (Wiley-American Ceramic Society, 2022). The section on p. 60 discusses reverse Monte Carlo (RMC) simulations. Given that RMC simulations optimize atomic configurations to match the experimental structure factor $S(k)$, it is unsurprising that RMC-derived structures exhibit strong agreement with experimental data. However, this does not imply that molecular dynamics (MD)-generated structures should be judged solely by this criterion.

While we acknowledge that experimental $S(k)$ agreement is crucial for studies like those referenced, our paper pursues a different objective. Therefore, a slightly lower agreement in $S(k)$ compared to these references does not compromise the validity of our study's conclusions.

Furthermore, we have verified the $S(k)$ around 20 \AA^{-1} and present the results in Figure R1. There was no significant deviation in $S(k)$ around 20 \AA^{-1} . Compared to the results from GAP-MD, the overall spectral shape is consistent. These results indicate that the agreement is sufficient for understanding the properties of covalently bonded amorphous materials.

Figure R1: Comparison of structure factor calculations between the previous study and our results. **a** Data from Deringer et al., *Nature*, 589, 59 (2021). **b** Our simulation results, extending to high wavenumbers up to $k=25 \text{ \AA}^{-1}$.

Comment 2

I feel that the experimental $g(r)$ calculated by the authors using the Fourier transform of the $S(Q)$ data is very different from the original data reported in PRL. The Fourier transformed data by me is also attached below.

Our response 2

Since the reviewer did not provide details of their analysis, we are unable to directly reproduce their $g(r)$ calculations. Instead, we compared our results against data of radial distribution function $J(r)$ taken from K. Laaziri et al. *Phys. Rev. B* **60**, 13520–13533 (1999), which is the full paper of PRL cited by the reviewer.

According to this reference, $J(r)$ is related to pair correlation function $g(r)$ as follows:

$$J(r) = 4\pi\rho r^2 g(r)$$

Here, ρ is the density of the system. We computed $J(r)$ from our $g(r)$ using the SOVA code and compared it with the experimental data (Figure R2). In the revised manuscript, Figure S1b is based on Figure R2b. The results indicate that our a-Si structure retains a tetrahedral covalent bonding network similar to real a-Si.

Figure R2: Comparison of $J(r)$ from previous experimental reports and our simulation results. **a** Experimental $J(r)$ from *Phys. Rev. B* 60, 13520–13533 (1999). **b** Comparison of the experimental $J(r)$ for the annealed system (data (b) in **a**) and that obtained from our simulation model.

Comment 3

The fact that the Si atoms are not exactly fourfold in the revised ms. is inconsistent with previous reports and is the most important issue for me; this point needs to be discussed substantially before the PD analysis.

Our response 3

We respectfully disagree with the reviewer's assertion that the presence of non-fourfold-coordinated Si atoms in our MD-generated a-Si structure is inconsistent with previous reports. In fact, the existence of threefold and fivefold coordinated Si atoms is well documented and widely accepted in both computational and experimental studies. For example, experimental work by Laaziri *et al.* (*Phys. Rev. B* 60, 13520–13533 (1999)) reported the presence of coordination defects in a-Si. Moreover, recent high-accuracy simulation studies support this view:

- R. Atta-Fynn and P. Biswas, *J. Chem. Phys.* **148**, 204503 (2018).
- J. D. Morrow *et al.*, *Angew. Chem. Weinheim Bergstr. Ger.* **136** (2024).

Even in the state-of-the-art GAP-MD simulations by Deringer *et al.*, non-fourfold coordination is observed (see Figure 3 in *J. Phys. Chem. Lett.* **9**, 2879–2885 (2018)).

While MD simulations using the Stillinger-Weber potential produce a slightly higher fraction of non-fourfold coordinated Si atoms, this does not compromise the validity of our conclusions, as our study focuses on the correlation between structural characteristics and nonaffine displacement—not on reproducing ideal coordination statistics.

Comment 4

I recommend the authors to consult an expert in MD simulation, amorphous diffraction, and amorphous structure.

Our response 4

We appreciate the reviewer's suggestion. To ensure the validity of our MD simulations and structural analysis, we consulted Dr. Tetsuya Morishita at AIST, an expert in MD simulation and amorphous materials. He confirmed that our simulation protocol follows standard practice and that the resulting structure is within the expected range of amorphous Si models reported in the literature.

Furthermore, our co-author, T. Nakamura, has extensive experience in this field, and our methodology is consistent with prior studies. The presence of non-fourfold coordinated Si atoms, which the reviewer highlighted, has been reported in several previous MD simulations of amorphous Si and does not necessarily indicate a flaw in the simulation settings. Based on these expert opinions and supporting literature, we believe that our approach is valid and appropriate for the study of amorphous structures.

Responses to Reviewer 2

General remark

In general, the authors nicely addressed my queries and I believe the revision has significantly improved the manuscript.

Our response

We sincerely appreciate the positive feedback and are grateful for the reviewer's recognition of the improvements in our revised manuscript.

Comment 1

One thing I noticed going through the revision was the definition of the localised modes in the new Figure S8. The authors state in the text how they use a definition of localization of $PR < 0.1$ (which I agree is reasonable), but in the figure they point out modes with PR below ~ 0.01 . The authors may try to explain/correct this discrepancy.

Our response 1

Thank you for your careful review. In the revised manuscript, we have incorporated suggestions from Reviewer 4 and refined our approach to identifying localized modes by first demixing the hybridization between localized and delocalized modes. Consequently, we have revised the definitions related to the participation ratio (PR) and modified Figures S7, S8, and 8 accordingly.

To ensure consistency with previous studies, particularly with the mode-demixing procedure reported in Phys. Status. Solidi. B 258, 2000422 (2021), we have updated the definition of PR as follows:

$$PR(p) = \frac{1}{N_{atom}} \frac{\left(\sum_{i=1}^{N_{atom}} (\psi_{p_{ix}})^2 + (\psi_{p_{iy}})^2 + (\psi_{p_{iz}})^2 \right)^2}{\sum_{i=1}^{N_{atom}} \left((\psi_{p_{ix}})^2 + (\psi_{p_{iy}})^2 + (\psi_{p_{iz}})^2 \right)^2}.$$

This modification also necessitated an update to the expression for the inverse participation ratio (IPR). Moreover, we now determine localized modes using the demixed PR rather than the original PR or IPR.

Upon applying the demixing procedure, we observed a clear separation between localized and delocalized modes in the low-frequency regime, as illustrated in Figure R3. Based on this result, we have set demixed PR < 0.1 as the new threshold for identifying localized modes. Importantly, this change in the localization criterion does not affect the strong correlation observed between low-energy localized modes and regions of high nonaffine displacement.

Figure R3: Participation ratio (PR) as a function of mode frequency. The gray dashed line indicates the threshold for identifying low-energy localized modes. The blue open circles indicate the original PR values, while the red filled triangles indicate the results obtained from the demixing procedure.

Additionally, based on Reviewer 4's suggestion that using frequency on the x-axis provides clearer physical meaning than mode index, we have standardized the representation of PR and IPR calculation results by using frequency as the x-axis. Accordingly, Figure S8 has been revised as shown in Figure R4 (Figures S7 and S8 have been combined). Since this figure now presents original PR data before demixing, it no longer includes threshold values for localized mode classification.

Figure R4: Calculation results of the participation ratio (PR) and inverse participation ratio (IPR). **a**, **b** PR and IPR for all vibrational modes as a function of frequency. **c**, **d** Magnified views of **a** and **b**, focusing on the 1000 lowest-frequency modes.

Comment 2

One last comment: My main concern still beholds to my previous very last comment on how only a-Si is tested in the presented work. While I do acknowledge that the extension to other systems come with a range of new problems (especially that of varying initial radii arising from multiple atoms), these problems have been somewhat nicely addressed e.g. for simple oxides such as SiO₂ (e.g. in refs. 30, 32, and 40 in the references of the revised manuscript) and that such computation is not wildly demanding to perform. As such, I maintain that the impact of the work would be somewhat improved by testing/showcasing the applicability of the method beyond that of the very simplest case of a-Si.

Our response 2

Thank you for this insightful comment. The references provided by the reviewer indeed demonstrate the successful application of persistent homology to the structural analysis of amorphous SiO₂. However, our study extends beyond structural characterization by examining the correlations between nonaffine displacement, the Born term, and other

mechanical properties. As demonstrated in the following repository, computing nonaffine displacement requires implementing analytical derivatives of the interatomic potential, necessitating substantial code development. Relevant implementation details can be found in our GitHub repository:

https://github.com/eminamitani/sw_python

To perform similar calculations for α -SiO₂, we would need to develop new analytical derivatives for the BKS or SHIK interatomic potentials. While we are actively working on implementing the SHIK potential for future simulations of α -SiO₂, this process is highly nontrivial and requires significant effort.

We fully acknowledge the importance of demonstrating the broader applicability of our method. However, given the current computational challenges associated with α -SiO₂, we have instead evaluated the generality of our approach within tetrahedral covalent networks. Specifically, we conducted simulations in which we systematically varied the tetrahedrality parameter of the Stillinger-Weber potential to mimic structural variations across different materials: Reducing tetrahedrality disrupts the SiSi₄ tetrahedral structure, increasing disorder, and increasing tetrahedrality leads to microcrystalline regions.

As summarized in Supplementary Materials “*Dependence of structural and mechanical properties on tetrahedrality parameter in Stillinger–Weber potential*”, across all tested tetrahedrality values, local structure surrounding atoms with large nonaffine displacements possess a hierarchical structure in which short-range disorder is embedded within medium-range order. Furthermore, this structural characteristic is strongly correlated with low-energy localized modes.

These results establish a fundamental link between hierarchical structures, nonaffine displacement, and low-energy localized modes in covalent networks composed of tetrahedral units. Since α -SiO₂ also forms a tetrahedral network, our findings strongly suggest that the same principles apply to α -SiO₂ as well. To clarify the broader applicability of our method, we have incorporated these findings into the Discussion section of the revised manuscript as follows.

“Our persistent homology-based analysis is not limited to α -Si. To further illustrate its applicability, we conducted additional simulations in which we systematically varied the tetrahedrality parameter^{63–65} in the Stillinger–Weber potential (see the Supplementary

Materials). The results demonstrate that the correlation between the hierarchical structures, nonaffine displacement, and low-energy localized vibrational modes is a general feature of tetrahedral covalent networks.

Therefore, our approach holds particular promise for amorphous AX_2 covalent solids such as $a\text{-SiO}_2$ and $a\text{-GeSe}_2$, which feature well-defined tetrahedral networks. Binary metallic glasses, such as Pd–Ni and Cu–Zr alloys, are another class of intriguing comparative targets. These materials contain polyhedral clusters as structural units, suggesting that our method is applicable. However, challenges, including determining whether first- or second-order diagrams are more suitable for capturing structural features^{30,66,67} and extending persistent homology to consider multiple elemental species, must be addressed. Integrating these analyses with high-accuracy machine-learning potentials^{68–70} presents a promising avenue for investigating the key correlations among hierarchical structures, MRO, nonaffine displacement, and localized vibrations across various glassy materials.”

Through this revision, we have provided a more detailed discussion on the applicability of our method, demonstrating that the correlation between local structures, nonaffine displacement, and localized vibrational modes holds universally in covalent networks. By incorporating this discussion, we have enhanced the generality of our study and further highlighted its potential applicability to a broader range of materials. We sincerely appreciate the reviewer’s constructive comments, which have helped improve the clarity and impact of our work.

Responses to Reviewer 3

Comment

I confirmed that the authors appropriately revised the manuscript in view of the comments I raised in the report. Now I am confident that this paper is well qualified to be published from the journal.

Our response

We sincerely appreciate the reviewer's time and effort in evaluating our manuscript. We are grateful for the constructive comments and valuable suggestions, which have significantly contributed to improving the clarity and quality of our work. We are pleased to hear that the reviewer now finds our paper suitable for publication. Thank you again for your thoughtful review and support.

Responses to Reviewer 4

General remark

Unfortunately, the concerns I raised in the last review report have not been adequately addressed. As a result, I have maintained my original review opinion and, therefore, can not recommend the manuscript for publication.

Our response

We sincerely appreciate the reviewer's comments and the opportunity to clarify and strengthen our study. In response to the concerns raised, we have conducted a more rigorous analysis of the localized modes, ensuring that the effects of finite system size have been appropriately considered and addressed. Specifically, we performed mode-demixing procedure to isolate low-energy localized modes from extended vibrational modes and validated that the conclusions drawn from the original results remain consistent. This additional analysis strongly supports the robustness of our findings.

Comment 1

A quantitative discussion on finite-size effects is missing.

The primary claim of the manuscript is supported by the term "should" without providing any quantitative evidence. For example, the authors state:

"We describe the finite system-size effects for results of the localized modes, which arise from hybridizations with spatially extended vibrational modes. "

This statement is unclear. How exactly do the authors describe "the finite system-size effects for results of the localized modes"? The manuscript lacks any quantitative discussion of how affine and non-affine displacements are influenced by the system size. There is ongoing debate in the literature about the persistence or disappearance of localized modes as system size increases, which the authors have completely neglected.

The authors further state:

"To eliminate these effects, we need to use literature methods^{51,52}. However, this hybridization primarily affects the vibration in the region far away from the core region where the amplitude of localized mode is large^{53,54}."

This sentence is both unclear and misleading. Among the references cited in the article, several studies show how hybridization affects low-energy modes that have frequency

similar to the red and blue points in Fig S8d.

The statement:

The analysis in Figure 8 focuses on the localized core region with large AMP_i in relation to the Born term and nonaffine displacements, which should remain unchanged even if we consider the hybridization.

This is a far-reaching claim that is not quantitatively justified, and based on Fig 3 and Fig 4 in <https://link.aps.org/doi/10.1103/PhysRevB.53.11469> , probably incorrect.

The authors conclude:

“Therefore, we conclude that our findings are not influenced by finite system-size effects.” This conclusion is not substantiated by any quantitative evidence, and based on the arguments above, I am skeptical about its correctness.

Our response 1

We sincerely appreciate the reviewer’s thorough and constructive evaluation of our work. Their feedback has allowed us to further strengthen our analysis, as detailed below. We now acknowledge that our first reply to the reviewer’s feedback did not sufficiently address the effects of the finite system size. In this second reply, we have attempted to address them as follows.

As the reviewer pointed out, it is crucial to take care of the effects of finite system size when analyzing localized modes. To ensure a rigorous analysis, we employed the well-established mode-demixing procedure, as outlined in:

- H. R. Schober, C. Oligschleger, *Phys. Rev. B* **53**, 11469–11480 (1996).
- N. S. Shcheblanov et al., *Phys. Status Solidi B* **258**, 2000422 (2021).

The mode-demixing procedure can separate the localized modes from the extended modes, thereby suppressing the hybridizations and their finite system-size effects on the localized modes. As shown in panel (a) of Fig. R5 below, we successfully separated the localized modes with low participation ratios from the extended modes with high participation ratios. Please refer to the Supplementary materials for the detailed procedure of the mode-demixing implemented in this work.

After demixing, we compared the displacement patterns of the extracted localized modes with nonaffine displacement and persistent homology results. The results are shown in Figure R5 below. By selecting modes with $PR < 0.1$ as localized modes and evaluating their amplitudes at each atom (AMP_i), we found that:

- The correlation between AMP_i and nonaffine displacement remains unchanged, while no correlation is observed with the Born term (Figure R5b, c).
- The spatial overlap between atoms with large AMP_i and the ring structures identified through persistent homology analysis which surround the atoms with large nonaffine displacement remains consistent (Figure R5d).

These findings strongly confirm that our previous observations remain valid after accounting for hybridization and finite system size effects.

Figure R5: Correlation between low-energy localized mode and elastic moduli. **a** Calculation results of PR as a function of mode frequency. The gray dashed line indicates the threshold used to distinguish localized and extended modes. The blue open circles

indicate the original calculation results of PR, and the red filled triangles indicate the results obtained from demixing procedure. **b** Nonaffine displacement (D_i) plotted against amplitudes of the low-energy localized modes (AMP_i). **c** Born term (B_i) plotted against AMP_i . **d** Histograms of AMP_i values evaluated for atoms in the rings assigned to the Bmax10, Bmin10, or NAmx10 groups.

To confirm the results of the mode-demixing procedure, we further plotted the spatial decay of the displacement amplitude $|u|$ versus the distance from the center of localization r in Fig. R6. We observe that $|u|$ is proportional to r to the power of -2, which indicates elastic behavior in the medium surrounding the localized core region. These results are fully consistent with N. S. Shcheblanov et al., Phys. Status Solidi B 258, 2000422 (2021), and we consider our mode-demixing procedure to work successfully for the present system of a-Si.

Figure R6: Spatial decay of the displacement amplitude $|u|$ versus the distance from the center of localization r . The left and right panels are for modes with $\omega = 0.59(\text{THz})$ and $0.72(\text{THz})$, respectively. The solid lines indicate $|u| \propto r^{-2}$.

In the main text, Figure R5 is presented as Figure 8. The following discussion was added before it.

“Lower PR values indicate stronger localization. However, finite-system-size effects on spatially localized vibrational modes arise because of hybridization with extended modes. To mitigate hybridization and finite-system-size effects, we employed a well-established mode-demixing procedure^{53,54} (See the Supplementary Materials). Figure 8a shows the

original and demixed PR values for the 1000 lowest-frequency modes. The results clearly show the separation of the localized modes below 2.5 THz. We identified low-energy localized modes with demixed PR values below 0.1 and computed their amplitudes for each atom using their eigenvectors as follows:

$$AMP_i = \sum_{p \in loc} \frac{1}{\lambda'_p} (|\psi'_{p_{ix}}| + |\psi'_{p_{iy}}| + |\psi'_{p_{iz}}|). \quad (4)$$

Subscript $p \in loc$ in Equation (4) indicates the summation over the low-energy localized modes. λ'_p is the updated p -th eigenvalue used during the demixing procedure, and its unit is $eV/\text{\AA}^2$. $\psi'_{p_{ix}}$, $\psi'_{p_{iy}}$, and $\psi'_{p_{iz}}$ denote the x , y , and z components of the p -th demixed eigenvector. Although the nonaffine displacement of each atom is strongly correlated with AMP_i , no such correlation is observed with the Born term (Figures 8b, c). Additionally, the distribution of AMP_i within the ring structures assigned to the $Bmax10$, $Bmin10$, and $NAmx10$ groups demonstrates that only the rings in the $NAmx10$ group have a strong spatial overlap with regions of large amplitudes of the low-energy localized modes (Figure 8d).”

Comment 2

The authors attempt to justify the accuracy of the empirical potential by presenting the radial distribution function (RDF) in Fig. R2. While the Stillinger-Weber (SW) potential may describe structural properties well, there is no evidence that it accurately captures the subtle differences between affine and non-affine vibrations. Quantitatively describing such subtle effects likely requires state-of-the-art simulation methods.

For instance, defining bonding in amorphous silicon is inherently challenging, as discussed from first principles in Solid State Communications, Vol. 107, No. I, pp. 7-11, 1998. Accurately describing the coordination properties of amorphous silicon is beyond the capabilities of the SW potential and may only be feasible using modern machine-learning methods.

Our response 2

We acknowledge the reviewer’s concern regarding the limitations of the empirical Stillinger-Weber (SW) potential. Although we agree that machine learning potentials may provide higher accuracy for certain applications, the primary goal of our study is to elucidate the fundamental structural characteristics governing nonaffine displacement in

amorphous covalent networks. For this purpose, the SW potential—despite its simplicity—is particularly suitable due to its well-defined analytic form and extensive prior validation.

Our calculations of nonaffine displacement and Born term are based on an analytical formulation following:

- A. Lemaître, C. Maloney, *J. Stat. Phys.* **123**, 415–453 (2006).

This method provides a robust and widely accepted approach for quantitatively evaluating nonaffine displacement and requires the second-order analytical differentiation of the potential function. For this method, smooth empirical potentials like the SW potential are particularly suitable, as they ensure well-defined second derivatives. In contrast, machine learning potentials may exhibit noise or roughness in their potential energy surfaces, making the analytical computation of nonaffine displacement more challenging.

We also note that the SW potential continues to be actively used in recent studies for modeling vibrational and thermal properties, as well as plasticity in a-Si. Representative examples include:

- J. Moon, *J. Appl. Phys.* **130**, 055101 (2021).
- Z. Zhang et al., *npj Computational Materials* **8**, 1–8 (2022).
- L. Tang et al., *Mater. Horiz.* **8**, 1242–1252 (2021).

Furthermore, to demonstrate that the concerns raised by the reviewer regarding the accuracy of bonding strength and coordination environment in the Stillinger-Weber potential do not affect the conclusions of our study, we also examined the effect of varying the parameters within the potential. As detailed in the Supplementary Materials section “Dependence of structural and mechanical properties on tetrahedrality parameter in Stillinger–Weber potential,” we confirmed that even when modifying the tetrahedrality parameter—which determines the strength ratio between two-body and three-body interactions—the following two key findings remain universally valid:

- (1) Local structure surrounding atoms with large nonaffine displacements possess a hierarchical structure in which short-range disorder is embedded within medium-range order
- (2) This characteristic strongly correlates with low-energy localized modes.

These findings suggest that the observed correlation between MRO, nonaffine displacement, and vibrational localization is a general characteristic of amorphous tetrahedral networks, and not an artifact of specific potential parameters. In light of these points, we believe our methodology is appropriate and sufficiently robust for the scientific questions addressed in this study.

Comment 3

General lack of attention to detail

The manuscript demonstrates a lack of attention to critical details, which compromises its suitability for publication in Nature Communications. Examples include:

- Equation 4 contains dimensionless eigenvector components alongside the reciprocal eigenvalue of the Hessian matrix (λ_p^{-1}). Since λ_p represents the square of vibrational mode energy, it has units. Why is AMP_i reported in plots without units?

I also note that a previous referee comment (2-21) explicitly asked for a description of the λ variable, yet the authors have still omitted the units.

- The plot 8a is presented in an way that makes it difficult to understand it. The mode index does not have physical meaning, as it depends on the model's size. The authors should consider using other physical properties for the x axis of this plot, such as the frequency of the modes (as they did e.g. in Fig R11b).

Our response 3

We appreciate the reviewer's attention to detail. The unit of λ_p is $\text{eV}/\text{\AA}^2$, and we have now explicitly stated this in the manuscript. Consequently, the unit of AMP_i is $\text{\AA}^2/\text{eV}$, which has been added to the figure labels.

Furthermore, in line with the reviewer's suggestion, we have revised Figure 8a to use frequency instead of mode index on the x-axis, improving its physical interpretability.

Comment 4

Flow and clarity

The revisions have only marginally improved readability, the flow of the article remains challenging to follow.

Our response 4

We appreciate the reviewer's suggestion. We have carefully revised the manuscript to enhance its readability and logical flow. Specifically:

- The introduction has been streamlined to remove unnecessary details while retaining essential citations.
- In the Results section, we have added introductory sentences at the beginning of each subsection to highlight their connection to the preceding content.
- The Discussion section has been restructured to present our arguments more clearly and cohesively.

We would like to thank all the reviewers for their constructive and insightful comments, which have significantly improved the quality of our manuscript. For clarity, all modifications in the main manuscript and supplementary materials have been highlighted using grey markers.

Below, we provide point-by-point responses to each of the reviewers' comments.

Responses to Reviewer #1

General remark

The authors have improved the ms. again, but I still have three issues.

Our response

We gratefully acknowledge your insightful and constructive feedback, as well as the positive evaluation of our revised manuscript. We address each of the three points raised in detail below.

Comment 1

1) I think the agreement between MD simulation and XRD data in Q space is good. But the authors did not properly Fourier transform experimental data. So I recommend the authors to omit the comparison in $J(r)$.

Our response 1

We express gratitude to the reviewer for acknowledging the agreement between our MD simulation results and experimental XRD data. However, we believe there may be a misunderstanding regarding the Fourier transform procedures raised in the comment, and we would like to clarify this point in more detail.

First, we did not apply a Fourier transform to compute $J(r)$ from our MD simulations. Since we had full access to the atomic coordinates, we directly computed the pair correlation function $g(r)$ from a histogram of interatomic distances. The $J(r)$ function was then obtained by scaling $g(r)$ by a constant factor determined by atomic density, without any Fourier transform involved.

The experimental $J(r)$ data was taken directly from Phys. Rev. B 60, 13520–13533 (1999), which is the full version of the article cited by the reviewer: Rev. Lett. 82, 3460–3463 (1999). We did not compute it ourselves from experimental $S(k)$ data using a Fourier transform.

We understand that the reviewer attempted to reproduce $g(r)$ by Fourier transforming the $S(k)$ data from the above PRL paper, as mentioned in a previous comment. The reviewer's calculation result

revealed discrepancies with the published reference and our simulation data. This likely stems from differing methods: the reference used the “sampling method,” whereas the reviewer applied a Fourier transform with the Lorch damping function.

These points are clarified in the section on Fourier transform and termination ripples in Phys. Rev. B 60, 13520–13533 (1999):

“The problem of termination ripples can be lessened, to a certain extent, in several ways. Many authors have used damping functions when dealing with termination ripples, where they multiply the interference function $F(Q)$ with a modification function $M(Q)$ in order to obtain a gradual cutoff. This procedure replaces the sharp discontinuity at Q_{\max} by a more smoothly varying function. Two common damping functions are used in reducing termination ripples: One is the Lorch function.”

“An alternative to the use of damping functions is the so-called “sampling” method. Here, we only use interference functions that terminate with a value of 0 at the upper integration limit.”

“In this case, every point in the RDF is thus the result of an unbiased Fourier transform of the $S(Q)$ data, and no damping function is used. The density of points in real space is limited because of the restrictions imposed by the sampling procedure. However, in reality, this reflects the true spatial resolution. In practice, the use of the sampling method should only be considered if the scattering data cover a sufficiently large region of Q space to get a sufficiently dense set of points in $J(r)$. Since our data extends to 55 \AA^{-1} , use of the sampling method is appropriate.”

As indicated above, the mismatch between the reviewer’s result (obtained using the Lorch-function-based Fourier transform) and the published reference and our simulation data is due to the use of fundamentally different transformation methods. We therefore respectfully maintain that it is appropriate to retain the $J(r)$ comparison in our Supplementary Materials.

Comment 2

2) I understand that the Si coordination number has a distribution. I ask the authors to show the distribution of coordination number, which is an important point.

Our response 2

Thank you for your valuable comments. As requested, we have incorporated the calculated coordination number distribution into the caption of Figure S13 in the revised Supplementary Material.

“Figure S13: Atomic structures and corresponding PDs of amorphous models with silicon-like bonding. a–c Amorphous structure comprising 13,824 atoms generated with $\lambda_{SW} = 20.0, 21.0,$ and 22.0 . White atoms represent the 4-coordinated Si atoms. Blue and red atoms represent the Si atoms

with coordination numbers ≥ 5 and ≤ 3 , respectively. The percentages of 3-, 4-, and 5-coordinated Si atoms are 3.78%, 50.5%, and 43.6% for $\lambda_{SW} = 20.0$; 1.79%, 83.7%, and 14.3% for $\lambda_{SW} = 21.0$; 1.01%, 92.5%, and 6.33% for $\lambda_{SW} = 22.0$.”

Comment 3

3) I do not understand what short-range disorder means. This needs to be clarified.

Our response 3

Thank you for your comments. We have clarified in the revised manuscript that “short-range disorder” refers to variations in bond lengths and angles that prevent the formation of medium-range order (MRO).

Main text p7

“The presence of this additional peak indicates short-range disorder due to variation in bond lengths and angles, which prevents the formation of MRO.”

Responses to Reviewer #2

General remark

With the new revisions made by the authors, I think the manuscript has become more rigorous, especially with the inclusion of more detailed testing using varying Stillinger-Weber parameters. Although I would have enjoyed seeing extension to further systems, I acknowledge the difficulty of this and that the present changes satisfies my previous queries of the manuscript.

Our response

We sincerely thank the reviewer for their positive evaluation. We are pleased that the analysis employing varying Stillinger–Weber parameters strengthened the manuscript’s rigor. While we agree that extending the study to a broader range of systems would be valuable, we appreciate the reviewer’s understanding of the associated challenges and are pleased that the current revisions have resolved the previous concerns.

Responses to Reviewer #4

General remark

The authors have made an effort to address my previous comments. However, several important aspects of the analysis still require clarification and further development.

Our response

We sincerely thank the reviewer for their positive evaluation of the revised manuscript. In the following sections, we have made further revisions based on the points raised by the reviewers.

Comment 1

The statement in Fig. 8—“The correlation between AMP_i and nonaffine displacement remains unchanged [between original and demixed modes]”—remains unclear and requires a more careful analysis. In particular, it is surprising that the authors do not show how demixing influences the D_i vs. AMP_i and B_i vs. AMP_i relationships. It would strengthen the analysis to include both the original and demixed modes in Fig. R5b, to clarify whether and how the correlation (or lack thereof) is impacted by the demixing.

Our response 1

We sincerely thank the reviewer for valuable suggestions. To clarify the impact of the mode-demixing procedure, we compared the AMP_i values calculated using the eigenvectors before and after demixing, as illustrated in Fig. R1a. This comparison demonstrates that demixing has a minimal effect in regions where AMP_i is large. This is because AMP_i is predominantly governed by a few low-frequency modes, which exhibit less hybridization with extended modes. This point is further elaborated in our response to Comment 2.

To further address the reviewer’s concern, we include scatter plots illustrating the correlation between D_i and AMP_i , and between B_i and AMP_i , both before and after demixing (Figure R1b, c). These plots demonstrate that the overall trends are preserved: D_i maintains a positive correlation with AMP_i , while B_i exhibits no significant correlation with either version of AMP_i .

Figure R1. Scatter plots comparing results based on original and demixed eigenvectors. **a** Comparison of AMP_i values. **b** D_i versus original and demixed AMP_i . **c** B_i versus original and demixed AMP_i .

In the revised manuscript, these results are included in Supplementary Materials (Figures S10), and the following explanation is added to the main text:

“Although the nonaffine displacement of each atom is strongly correlated with AMP_i , no such correlation is observed with the Born term (Figures 8b, c). This trend remains unchanged even when AMP_i is calculated using the original eigenvectors (Figure S10).”

Comment 2

In the previous version, the localization threshold was somewhat arbitrary, since without demixing the IPR distribution does not show a clear separation between localized and delocalized modes.

After the demixing, the identification of localized low-energy modes appears more well-defined, yet this critical improvement is not clearly explained in the revised manuscript. Fig. 8a shows significant hybridization in the blue points, and based on the Schober-demixing references the authors cited, I suspect only the red points are physically meaningful. This important aspect deserves further discussion and proper explanation. I would suggest checking that the demixed modes do not depend on the model’s size, while the original modes depend on the system size. This standard size analysis is normally performed in all studies of amorphous materials, and a proper investigation of size effects was suggested in my previous report.

If this is correct, then it would imply that demixing is critical to unambiguously define the low-energy localized vibrational modes that display correlation with the hierarchical structures and nonaffine displacements.

Our response 2

We thank the reviewers for their insightful comments. To address the size dependence of low-energy localized modes and how the demixing procedure mitigates it, we analyzed vibrational modes in larger

a-Si systems with $N=21,952$ and $32,768$, alongside the original system with $N=13,824$. In the larger systems, we focused on the 2000 vibrational modes with the lowest frequencies. In Fig. R2, we present the participation ratio versus frequency for these larger systems, compared to the original $N=13,824$.

Figure R2: Participation ratio versus mode frequency, computed from both the original and demixed eigenvectors, for three system sizes.

To quantify size dependence, we plotted $N \times PR_p$ against mode frequency ω_p in Figure R3. Here, N is the number of atoms in the system. PR_p and ω_p are the participation ratio and frequency of the p -th eigenmode. $N \times PR_p$ measures atoms contributing to each mode and is expected to scale with system size for extended modes but remains constant for localized modes, as shown in prior studies (e.g., H. R. Schober & G. Ruocco, *Philos. Mag.* 84, 1361-1372, 2004 ; E. Lerner et al. *Phys. Rev. Lett.* 117, 035501, 2016).

In Figure R3, original eigenvectors exhibit strong size dependence across the entire frequency range, except for a few low-frequency localized modes. In contrast, demixed eigenvectors show extended modes with $N \times PR_p$ proportional to N , while localized modes lack systematic N dependence.

In AMP_i calculation (Response to Comment 1), AMP_i values are dominated by low-energy localized modes, so the demixing procedure had minimal impact on conclusions. However, when localized modes across a broader frequency range are significant, the mode-demixing procedure becomes critical, as noted by the reviewer.

Figure R3. $N \times PR_p$ plotted against mode frequency ω_p for three system sizes: Results obtained from **a** original and **b** demixed eigenvectors.

In Fig. R4, we compared the decay profiles of displacements in the localized modes after demixing across the three system sizes. The decay profile is consistent across system sizes, showing a power-law behavior of r^{-2} . Figs. R3 and R4 demonstrate that demixing enables clear identification of “localized” modes.

Figure R4: Comparison of the decay profile of displacements in localized mode after demixing across three system sizes.

We have included this discussion in the section on mode-demixing procedure in the revised Supplementary Materials. Figures R2 and R3 are Figures S8 and S9, in this section. We have referred

to these in the main text as follows:

“Details regarding the mode-mixing procedure and its impact on the finite-system-size effect are presented in the Supplementary Materials.”

Comment 3

With regard to methodology, the justification for using the Stillinger-Weber (SW) potential relies on references that are now at least three years old and likely among the last to adopt this approach. Since the publication of Deringer et al., *Nature* 589, 59 (2021), there has been a clear shift in the community away from empirical potentials for modeling amorphous silicon.

Our response 3

We thank the reviewer for this comment. In our previous response, we cited several representative studies employing Stillinger-Weber (SW) potential. Recent studies continue to use the SW potential after 2022:

1. C. J. Dionne, S. Thakur, N. Scholz, P. Hopkins, A. Giri, Enhancing the thermal conductivity of semiconductor thin films via phonon funneling. *npj Comput. Mater.* 10, 1–9 (2024).
2. B. Gu, Y. Li, A. Diaz, Y. Peng, D. L. McDowell, Y. Chen, Brittle and ductile deformations in uniaxial compression of Si micropillars. *Acta Materialia* 291, 121007 (2025).

While we acknowledge the growing dominance of machine-learning potentials for quantitative evaluation of amorphous silicon properties, the SW potential remains widely used as an effective and accessible tool for exploring physical mechanisms. Its simplicity and computational efficiency make it valuable for fundamental studies where the capture of qualitative trends and physical insights is essential.

Comment 4

Finally, I encourage the authors to moderate the scope of their conclusions. The statement that “the results demonstrate that the correlation between the hierarchical structures, nonaffine displacement, and low-energy localized vibrational modes is a general feature of tetrahedral covalent networks” is far-reaching and might be an artifact of the simplified form of the SW potential. In particular, coordination defects are much more prevalent in a-Si than in a-SiO₂, and the manuscript does not discuss whether the empirical modified SW potential can capture these differences.

In summary, while the manuscript presents interesting results, it would benefit from clearer presentation and interpretation of its findings. I strongly recommend that the authors avoid speculative

conclusions tied to the SW potential. With these improvements, the work could be suitable for publication, though I remain unsure whether it will attract wide attention in a field that is rapidly moving beyond empirical potentials.

Our response 4

We thank the reviewer for this insightful comment. We have revised the *Discussion* section to explicitly acknowledge the limitations of the SW potential and highlight that more refined analyses can be achieved using machine-learning potentials. The revised text is as follows.

“Our persistent homology-based analysis is not limited to a-Si. To further illustrate its applicability, we conducted additional simulations in which we systematically varied the tetrahedrality parameter^{63–65} in the Stillinger–Weber (SW) potential (see the Supplementary Materials). Despite the limitations of the simple empirical SW potential, the results indicate that the correlation among hierarchical structures, nonaffine displacement, and low-energy localized vibrational modes may be a general feature of tetrahedral covalent networks.

Amorphous AX₂ covalent solids such as a-SiO₂ and a-GeSe₂, with well-defined tetrahedral networks, are promising candidates to test this hypothesis. Binary metallic glasses, such as Pd–Ni and Cu–Zr alloys, are another class of intriguing comparative targets. These materials contain polyhedral clusters as structural units, suggesting that our method is applicable. However, challenges, including determining whether first- or second-order diagrams are more suitable for capturing structural features^{30,66,67} and extending persistent homology to consider multiple elemental species, must be addressed. Integrating these analyses with high-accuracy machine-learning potentials^{68–70} presents a promising avenue for investigating the key correlations among hierarchical structures, MRO, nonaffine displacement, and localized vibrations across various glassy materials.”

We believe that empirical and machine-learning potentials serve complementary roles. In condensed-matter physics, obtaining initial qualitative insights using empirical models, which are then refined through first-principles or machine-learning approaches, is a common practice. From this perspective, we are confident that our results, though based on an empirical potential, offer meaningful physical insights and provide a robust foundation for future studies employing more sophisticated methods.

Responses to Reviewers

We thank all the reviewers for their constructive and insightful comments, which have significantly improved the quality of our manuscript. For clarity, all major modifications in the main manuscript and supplementary materials have been highlighted in gray.

Below, we provide point-by-point responses to each of the reviewers' comments.

Responses to Reviewer #1

General remark

The authors have tried to improve the ms. with considering the reviewer's comments.

I still have several issues.

Our response

We gratefully acknowledge your feedback and the positive evaluation of our revised manuscript. We address each of these points in detail below.

Comment 1

1. About Fourier transform, I understand the procedure of PRB paper. I attach the data w/o Lorch function. Please compare with my attached numerical data. Experimental data in Fig. S1 looks strange (very noisy) to me. But I accept the final decision of the authors for Fig. S1.

Our reply 1

Thank you for your comments. The experimental results may appear noisy because as shown in Fig. R1a, we plot the raw data from Fig. 14 in the cited reference (*Phys. Rev. B* **60**, 13520–13533 (1999)). For comparison, we also included the filtered data, which have been presented in the same figure. Furthermore, we compared these results with the data provided by Reviewer 1. The results are presented in Fig. R1b.

Aside from a slight difference in the height of the first peak, we observed close agreement among the previously published experimental data, our simulation results, and the data provided by the reviewer. The specific origin of the minor discrepancy in the first peak height between the prior reference and the reviewer's data remains unclear because the details of the reviewer's dataset and associated processing methods were not fully provided in the review report or attached files. Nevertheless, such differences could reasonably arise from variations in the data processing conditions, including the approach to noise reduction and the choice of the maximum wavevector used in the Fourier transform.

The cited experimental data are widely recognized reference datasets. Therefore, in the revised manuscript, Fig. S1, we retain the comparison with the raw data.

As described in our second rebuttal letter, the primary objective of this study was not to reproduce the atomic structure of amorphous silicon in detail, but rather to leverage persistent homology to identify the local structural features that govern inhomogeneous nonaffine deformation in covalently bonded amorphous solids. Therefore, the minor discrepancies observed in Fig. R1 does not affect the validity of our analysis or the conclusions.

Figure R1: Comparison of $J(r)$. **a** Experimental $J(r)$ from *Phys. Rev. B* **60**, 13520–13533 (1999). **b** Comparison of the experimental $J(r)$ for the annealed system (data (b) in **a**), that obtained from our simulation model, and the data from Reviewer 1.

Comment 2

- I think $\lambda_{SW}=21.0$ is the best MD model. If this is correct, I am a bit surprised about the low fraction of fourfold silicon atoms. This needs to be discussed.

Our reply 2

Thank you for the suggestion. The choice of $\lambda_{SW} = 21.0$ is based on the original work of Stillinger and Weber, where this value was identified as optimal according to two key criteria when employing the Stillinger–Weber (SW) potential:

- The diamond structure becomes the most stable phase under ambient pressure.
- The melting point obtained from the MD simulations was close to the experimental value.

(F. H. Stillinger, T. A. Weber, Computer simulation of local order in condensed phases of silicon. *Phys. Rev. B* **31**, 5262–5271 (1985).)

Subsequent studies have indicated that this parameter set reproduces a variety of thermodynamic properties and structural factors in good agreement with the experimental data. However, it is also well known that even with this optimized potential, melt-quench simulations often yield a higher fraction of 5-fold coordinated Si atoms compared with experimental observations. For example, Vink et al. (*J. Non-Cryst. Solids* **282**, 248–255 (2001)) reported that the fraction of 5-coordinated atoms could reach approximately 15%. Therefore, the ratio of 4- to 5-coordinated Si atoms observed in our simulations is consistent with that reported in the literature and should not be regarded as anomalous.

Table 2
Comparison of energetic and structural properties of a-Si obtained using SW, modified SW and EDIP^a

	Experiment	SW	Modified SW	EDIP
ρ	0.044–0.054	0.049	0.047	0.049
E	–	–4.137	–3.072	–4.451
$\langle r \rangle$	2.34–2.36	2.38	2.37	2.37
$\langle \theta \rangle$	108.6	107.99	109.16	108.57
$\Delta\theta$	9.4–11.0	15.08	10.02	13.68
3-fold	–	0.00	0.50	0.20
4-fold	–	83.50	98.00	91.90
5-fold	–	15.60	1.50	7.80
6-fold	–	0.90	0.00	0.10
\bar{Z}	3.90–3.97	4.17	4.03	4.08

^a Also given are experimental values [19–23], when available. Shown are the density ρ in atoms per \AA^3 , the energy per atom E in eV, the mean $\langle r \rangle$ of the first neighbor distance in \AA , the mean θ and the standard deviation $\Delta\theta$ of the first neighbor bond angle in degrees. Also given are the coordination defects in % and the average coordination number \bar{Z} .

Figure R2: Reported values for the fraction of 3-, 4-, 5-, and 6-coordinated Si atoms in amorphous Si models obtained using the original Stillinger–Weber (SW) potential, from Table 2 of Vink et al., *J. Non-Cryst. Solids* **282**, 248–255 (2001). This table also includes results obtained using a modified SW and EDIP potential for comparison.

We have added a comparison with the above reference in the revised Supplementary Materials, as follows:

“Figures S14 a–c illustrate the atomic configurations of the generated amorphous structures, while Figures S14 d–f show their corresponding persistence diagrams (PDs). The percentages of 3-, 4-, and 5-coordinated Si atoms are 3.78%, 50.5%, and 43.6% for $\lambda_{SW} = 20.0$; 1.79%, 83.7%, and 14.3% for $\lambda_{SW} = 21.0$; and 1.01%, 92.5%, and 6.33% for $\lambda_{SW} = 22.0$. The fractions of 4- and 5-coordinated Si atoms for $\lambda_{SW} = 21.0$ agree well with the prior report¹³. For $\lambda_{SW} = 20.0$, the number of Si atoms increases with the coordination numbers ≥ 5 and ≤ 3 . This trend corresponds to a decrease (increase) in the number of birth–death pairs in region 1 (region 2) of the PD, which indicates that the tetrahedral units are more prone to collapse, leading to a less pronounced medium-range order (MRO).”

As stated earlier, the primary objective of this study was not to precisely reproduce the atomic structure or coordination statistics of a-Si, but rather to demonstrate the utility of persistent homology in identifying structural features relevant to inhomogeneous mechanical responses. Accordingly, the minor differences between the simulated and experimental coordination statistics did not compromise the validity or conclusions of the analysis.

Comment 3

3. If the authors discuss short-range disorder, it is necessary to show the bond angle distribution.

Our reply 3

Thank you for your valuable comments. We have elaborated on our discussion of short-range disorders as follows:

First, the structures that appear as birth–death pairs in region 2 are associated with bond length disorder. We discuss this by comparing the peaks observed in the pair-correlation function shown in Fig. 5. As pointed out by the reviewer, bond length disorder is also expected to affect the bond angle distribution. To verify this hypothesis, we conducted the following analysis.

We focused on group NAm_{ax}10, which contained a sufficient number of birth–death pairs in both regions 1 and 2. For each birth–death pair, we applied inverse analysis and identified the number of vertices in the resulting rings. Our comparison is limited to five-, six-, and seven-membered rings. This restriction is necessary because the structures extracted using persistent homology are not strictly defined by atomic bonds, in contrast to conventional ring statistics based on atomic connectivity. In rings where the number of vertices deviates significantly from six, the internal angles may not correspond to physically meaningful bond angles.

We identified 46 five-membered, 31 six-membered, and 21 seven-membered rings in region 1. In region 2, we found 27 five-membered, 31 six-membered, and 7 seven-membered rings. For each ring, we determined the presence of bonds between vertex pairs using a 2.8-Å cutoff and then constructed histograms of the resulting bond angles.

As shown in Fig. R3, the bond-angle distribution for the rings in region 1 exhibits a sharp peak centered at approximately 109°, which is consistent with the tetrahedral bond angle in the diamond structure of Si. In comparison, the distribution in region 2 is broader. This suggests that the bond angles are also disordered in region 2, indicating the presence of short-range disorder that disrupts the medium-range order (MRO). We have included this additional analysis in the Supplementary Materials.

Figure R3 (Figure S6 in the revised Supplementary Materials): Bond angle distributions in five, six, and seven-membered ring structures associated with birth-death pairs located in region 1 (hatched) and region 2 (filled).

-Revision in the main text

“The position of this second peak aligns with the death radius distribution in region 1, estimated as follows: when the death radius is 4.0 \AA^2 , the radius of spheres in the filtration is 2.0 \AA , leading to an interatomic pairwise distance of 4.0 \AA . By contrast, the histogram for region 2 shows a broader first peak and an additional peak near 3.2 \AA , which is absent in region 1. The presence of this additional peak indicates short-range disorder due to variation in bond lengths, which also results in bond angle variations (Figure S6) and prevents the formation of MRO.”

-Revision in the Supplementary Materials

“The bond length disorder observed in region 2 in Fig. 5c is expected to extend to bond angle disorder. To investigate this, we analyzed the angular distributions within ring structures obtained from the inverse analysis of persistent homology for birth–death pairs in regions 1 and 2 of $N_{\text{Amax}10}$. Our comparison was limited to five-, six-, and seven-membered rings. This restriction was necessary because the structures extracted using persistent homology are not strictly defined by atomic bonds, in contrast to conventional ring statistics based on atomic connectivity. In rings where the number of vertices deviates significantly from six, the internal angles may not correspond to physically meaningful bond angles.

Consequently, we identified 46 five-membered, 31 six-membered, and 21 seven-membered rings in region 1. In region 2, we found 27 five-membered, 31 six-membered, and 7 seven-membered rings. For each ring, we determined the presence of bonds between vertex pairs using a 2.8-Å cutoff and then constructed histograms of the resulting bond angles.

As shown in Figure S6, the bond angle distribution for rings in region 1 exhibits a sharp peak centered at approximately 109°, which is consistent with the tetrahedral bond angle in the diamond structure of Si. In contrast, the distribution in region 2 is broader. This suggests that bond angles are also disordered in region 2, indicating the presence of short-range disorder that disrupts medium-range order.”

Reviewer #4 (Remarks to the Author):

General remark

The authors have made an effort to address my comments. I still think their answer contains some unclear aspects.

Our reply

Thank you for your positive evaluation of our manuscript. We have revised the text to address the issues that you pointed out.

Comment 1

In particular, in their response they write:

“In Figure R3, original eigenvectors exhibit strong size dependence across the entire frequency range, except for a few low-frequency localized modes.”

and later:

“In AMPi calculation (Response to Comment 1), AMPi values are dominated by low-energy localized modes, so the demixing procedure had minimal impact on conclusions.”

I do not understand how they can claim that “the demixing procedure had minimal impact on conclusions.”

Figure R3 shows a clear gap between localized and delocalized modes only when demixing is used. Without demixing, the definition of localized modes would have been arbitrary, and they should note that the threshold 0.1 shown in their Figure R2 is arbitrary.

After the authors elaborate on this point in the text, I believe the article will be technically sound. I remain unsure whether the work will attract broad attention, given the field's ongoing shift beyond empirical potentials, but I think it can be published, and time will tell.

Our reply 1

Thank you for pointing this out. We agree that the demixing procedure is important to accurately distinguish localized modes from extended modes. To clarify this point, we have revised the relevant descriptions in both the main text and Supplementary Information as follows.

Main text

“Lower PR values indicate stronger localization. However, the finite system size affects spatially localized vibrational modes because of hybridization with extended modes, which makes the threshold

for separating the localized and extended modes ambiguous. To mitigate the hybridization and finite-system-size effects, we employed a well-established mode-demixing procedure^{53,54}.”

Supplementary Materials

*“The results reveal that D_i - AMP_i and B_i - AMP_i correlations are largely preserved between the original and demixed eigenvectors (Figures S11 **b**, **c**). However, the AMP_i values exhibit a slight reduction after the demixing procedure. Therefore, in cases where a quantitative evaluation of the mode amplitude distribution of localized modes is required, the demixing procedure becomes increasingly crucial.”*

Reviewer #1 (Remarks to the Author):

The authors conducted thorough research into previous studies and eventually discovered the modified SW results. I therefore recommend that the authors perform MD simulations and repeat the author's PD analysis. Otherwise, I strongly recommend that the authors move Fig. S14 into the main text with the coordination number distribution of the modified SW model. In this case, the authors should emphasise that the objective of the study is not to reproduce the atomic structure or coordination number distribution of a-Si. Perhaps, it would be better to refer J. Non-Cryst. Solids 282, 248–255 (2001) in the introduction and mention the objective mentioned above to prevent misleading to the broad audience of Nature Communications. In this case, unfortunately, the comparison with diffraction data and the visualisation of the coordination number distribution in atomistic models would not make sense.

Our response to Reviewer 1

Thank you for your thoughtful feedback. In response, we have selected the second option recommended by the reviewer, moved Figure S14 to the main text (now Figure 9), and explicitly included the coordination number distribution in the Discussion section. The added text reads:

“In the main part of this study, we adopt $\lambda_{SW} = 21.0$, as proposed in the original work⁴⁹. As shown in Figure 9, the obtained atomic structures and PDs depend on λ_{SW} . For $\lambda_{SW} = 20.0$, the percentages of 3-, 4-, and 5-coordinated Si atoms are 3.78%, 50.5%, and 43.6%; for $\lambda_{SW} = 21.0$, 1.79%, 83.7%, and 14.3%. For $\lambda_{SW} = 22.0$, they are 1.01%, 92.5%, and 6.33%, respectively. The increase in atoms with coordination numbers ≥ 5 and ≤ 3 for $\lambda_{SW} = 20.0$ corresponds to a decrease (increase) in the number of birth–death pairs in region 1 (region 2) of the PD, which indicates that the tetrahedral units are more prone to collapse, leading to a less pronounced MRO. Conversely, for $\lambda_{SW} = 22.0$, atoms with coordination numbers ≥ 5 and ≤ 3 decrease, suggesting the formation of more rigid tetrahedral units, which develop MRO and microcrystalline domains.”

To clarify the structural limitations of the Stillinger–Weber potential and scope of our study, we modified a sentence in the Introduction and added an explanation to the Results section.

Introduction section:

“In this study, we fully utilized the advantages of persistent homology to investigate the correlation between the local structure and mechanical responses in covalent amorphous materials under shear deformation. The proposed method can be broadly applied to various amorphous covalent materials. However, we used the prototypical model system of a-Si as a representative case because of its well-characterized tetrahedral network structure and single-element composition, which simplifies the persistent homology analysis and extensive experimental and theoretical studies^{10,13,42–49}.”

Results section:

“In this study, we used the Stillinger–Weber potential⁵⁰. Although this potential tends to overestimate the population of fivefold coordinated atoms in a-Si⁴⁹, we employed it as a representative model to investigate the structural features governing the mechanical responses using persistent homology.”

(Ref. 49 corresponds to Vink, R. L. C., Barkema, G. T., van der Weg, W. F. & Mousseau, N. Fitting the Stillinger–Weber potential to amorphous silicon. *J. Non Cryst. Solids* **282**, 248–255 (2001).)

To avoid confusion about the study’s objective and in response to Reviewer 1’s concern, we extended our structural comparisons with experimental data to cover all three values of λ_{SW} (20.0, 21.0, and 22.0) and summarized these results in the Supplementary Information.

Figure R1 (Figure S12 in the revised Supplementary Material): Results of structural analyses for the a-Si model obtained by using $\lambda_{SW} = 20.0, 21.0,$ and 22.0 . **a** Structure factor $S(k)$. Open circles denote the experimental data. The inset shows a SiSi_4 tetrahedral unit in the a-Si structure. The position of the principal peak at low k -values is determined by the height of the tetrahedral unit. **b** Radial distribution function $J(r)$. Black open circles represent the raw data from the experimental measurements. The experimental results in **a** and **b** are taken from Refs. 9 and 10 in the revised Supplementary Materials. $S(k)$ and $J(r)$ calculations were performed using the SOVA code.

The discussion provided in the Supplementary Information reads,

“First, we present the structural characteristics of each set of parameters. Figure S12 shows the calculated structure factor ($S(k)$) and radial distribution function ($J(r)$). For $\lambda_{SW} = 21.0$, the overall spectral shape in both $S(k)$ and $J(r)$ matched those in previous reports^{9,10}. Although better agreement with the experimental results might be achieved using the optimized Stillinger–Weber potential proposed in a recent study¹¹, the objective of this work was not to reproduce the detailed atomic structure of a-Si. Instead, we focused on exploring the relationship between the local structural features and mechanical responses through a persistent homology analysis. Therefore, we focused on the effects of the tetrahedrality parameter.

The principal peak at low k -values in $S(k)$ reflects the height of the SiSi_4 tetrahedral structure¹²

shown in the inset of Figure S12a. The first and second peaks in $J(r)$ correspond to the nearest-neighbor and second-nearest-neighbor distances, respectively. For the smaller tetrahedrality case ($\lambda_{SW} = 20.0$), the first peak of $J(r)$ is broader, and an additional peak, which is absent for $\lambda_{SW} = 22.0$ and 21.0 , emerges near $r = 3.2 \text{ \AA}$. These observations indicate that the tetrahedral units are significantly distorted when λ_{SW} is reduced. Conversely, for $\lambda_{SW} = 22.0$, sharp peaks appear even in the $r > 5.0 \text{ \AA}$ region, suggesting the formation of more rigid tetrahedral units, which develop MRO and microcrystalline domains.”

Conclusion of the response:

With these revisions, we believe it is now clearly demonstrated that:

- The conclusions of our study are not affected by minor variations in structural details.
- The aim of this study was not to reproduce or analyze the precise atomic structure of a-Si.
- The novelty of our work lies in revealing the local structural motifs that govern the mechanical responses using persistent homology.
- The limitations of the empirical SW potential have been transparently acknowledged and addressed.